# The geography of climate and the global patterns of species diversity

Marco Túlio P. Coelho[1✉], Elisa Barreto[1], Thiago F. Rangel[2], José Alexandre F. Diniz-Filho[2], Rafael O. Wüest[1], Wilhelmine Bach[1,3], Alexander Skeels[1,3], Ian R. McFadden[1,3,4], David W. Roberts[5], Loïc Pellissier[1,3], Niklaus E. Zimmermann[1] & Catherine H. Graham[1]

Climate's effect on global biodiversity is typically viewed through the lens of temperature, humidity and resulting ecosystem productivity[1–6]. However, it is not known whether biodiversity depends solely on these climate conditions, or whether the size and fragmentation of these climates are also crucial. Here we shift the common perspective in global biodiversity studies, transitioning from geographic space to a climate-defined multidimensional space. Our findings suggest that larger and more isolated climate conditions tend to harbour higher diversity and species turnover among terrestrial tetrapods, encompassing more than 30,000 species. By considering both the characteristics of climate itself and its geographic attributes, we can explain almost 90% of the variation in global species richness. Half of the explanatory power (45%) may be attributed either to climate itself or to the geography of climate, suggesting a nuanced interplay between them. Our work evolves the conventional idea that larger climate regions, such as the tropics, host more species primarily because of their size[7,8]. Instead, we underscore the integral roles of both the geographic extent and degree of isolation of climates. This refined understanding presents a more intricate picture of biodiversity distribution, which can guide our approach to biodiversity conservation in an ever-changing world.

It is not clear why greater species richness is observed in warmer and more humid regions. Since the first description of large-scale patterns of species diversity by Alexander von Humboldt in 1807 (ref. 9), differences in climatic conditions between tropical and extratropical regions have been considered a plausible explanation for global diversity gradients. It is now well established that species richness is often strongly correlated with climatic conditions, namely temperature, water availability and resulting ecosystem productivity[1–6]. Although several causal pathways link climatic variables to species diversity including both current ecology and deep-time evolutionary processes[5,10], we lack an understanding of how the geography of climate is associated with diversity patterns. Specifically, it is unknown whether greater diversity in warmer and more humid regions results from the effect of the climate itself, or from the geographic area occupied by these climatic conditions (that is, climate area) and the spatial isolation between similar climatic conditions (that is, climate isolation), both of which could lead to higher diversity.

In an influential piece[7], John Terborgh argued that "unusual environmental situations will carry impoverished flora in relation to nearby sites incorporating more usual conditions". Conversely, more usual or common climatic conditions (that is, those that cover a greater geographic area) will have high species richness[7]. The relationship between climate area and species richness led to the area hypothesis to explain global patterns of species diversity, originally arguing that climatic conditions occurring in the tropics are more common than those conditions occurring in extratropical regions. The area hypothesis was further popularized and amplified by Michael Rosenzweig[8], but the emphasis on climate area was lost. At present, the area hypothesis is considered in a very broad geographic context building on the idea that tropical regions have more land than extratropical regions, thereby offering a reasonable explanation for why greater species richness is observed in the tropics. More recently, the commonness of climatic conditions (that is, the total global extent of a given climatic condition) has been used to explain empirical patterns of species richness[11], leading to the formalization of species–area relationship within climate[12]. In addition to area, the isolation of fragments of a given climatic condition on the surface of the Earth can also be expected to affect species diversity. Speciation rates are expected to increase as a result of reduced gene flow and increased environmental heterogeneity[13–15]. The same expectation applies when biodiversity is analysed within different climates. Climatic conditions with isolated and larger extent across the surface of the globe should reduce extinction rates, facilitate allopatric speciation and shelter biotas that evolved independently, and therefore strongly affect climatic gradients of diversity at the global scale. However, the combined effects of the geography of climate (that is, climate area and isolation) and climatic conditions per se on global patterns of diversity remain unknown.

[1]Swiss Federal Institute for Forest, Snow and Landscape Research, Birmensdorf, Switzerland. [2]Departamento de Ecologia, Universidade Federal de Goiás, Goiânia, Brazil. [3]Ecosystems and Landscape Evolution, Institute of Terrestrial Ecosystems, Department of Environmental System Science, ETH Zürich, Zurich, Switzerland. [4]Institute for Biodiversity and Ecosystem Dynamics, University of Amsterdam, Amsterdam, The Netherlands. [5]Ecology Department, Montana State University, Bozeman, MT, USA. ✉e-mail: marcotpcoelho@gmail.com

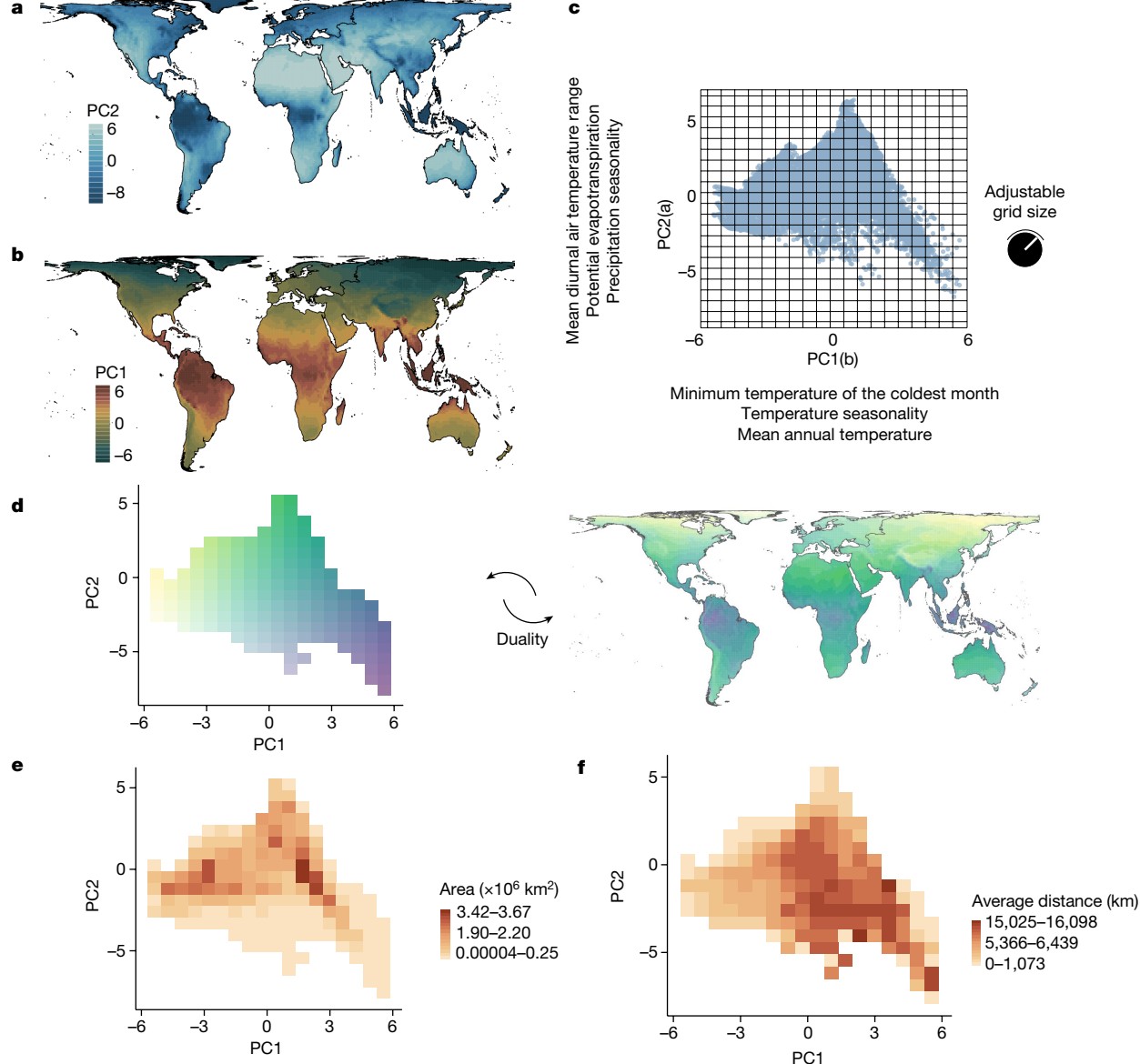

**Fig. 1 | From geographic to climate space. a,b**, Scores of the first and second principal components (PC1 and PC2) mapped across the world. **c**, Climatic information represented in a two-dimensional coordinate plane in which the $x$ axis and $y$ axis are represented by PC1 and PC2 and each point represents the climatic information of each geographic location ($n = 13,312$ geographic cells). **d**, Using the gridded climatic space created with PC1 and PC2 (**c**), the duality between climate and geographic spaces is shown by matching colours between climate and geographic grids, in which similarity in colour tone indicates similarity in climatic condition. The duality between geographic and climate space refers to the relationship between geographic and climate space: a given climatic condition (climate cell in climate space) is observed in several geographic locations, and several geographic locations belong to a unique climate condition. **e**, Climate area representing the sum of land surface area of a given climatic condition (that is, climate cell). **f**, Climate isolation representing the average geodesic distance among climate fragments (that is, geographic cells connected to each other that occur within a single climate).

A substantial challenge in assessing the impact of the geography of climate, specifically climate area and isolation, arises from the necessity to reorient traditional biodiversity studies—typically focused on geographic landscapes—towards the realm of climate space. This pivot demands a fresh perspective on examining the relationships between diversity and climate. Using multidimensional space, defined by climatic conditions, to study biodiversity patterns is not a new concept. It has been previously proposed in the literature[16–18]. Primarily, this approach has been used for the classification of life zones and biomes[19,20], as well as in species distribution modelling and its various applications[21,22]. By contrast, this perspective has been generally overlooked in studies investigating the emergence and maintenance of large-scale diversity patterns[11,12]. Here we investigate tetrapod diversity patterns thoroughly in climate space, disentangling the effects of the geography of climate and climate conditions per se, on diversity–climate relationships. We anticipate finding a higher number of species in climatic conditions that both cover large geographic areas and exhibit characteristics of isolation or fragmentation. We believe that these findings will hold true regardless of the inherent differences in thermal physiologies, such as endothermy and ectothermy, among tetrapod groups. These expectations are aligned with empirical evidence showing the effect of geographic area and isolation on geographic patterns of species richness independent of species' thermal and metabolic physiologies[8,13,15,23]. The shift towards understanding diversity as a pattern in climate space, driven by a process in climate space, can reveal insights into how

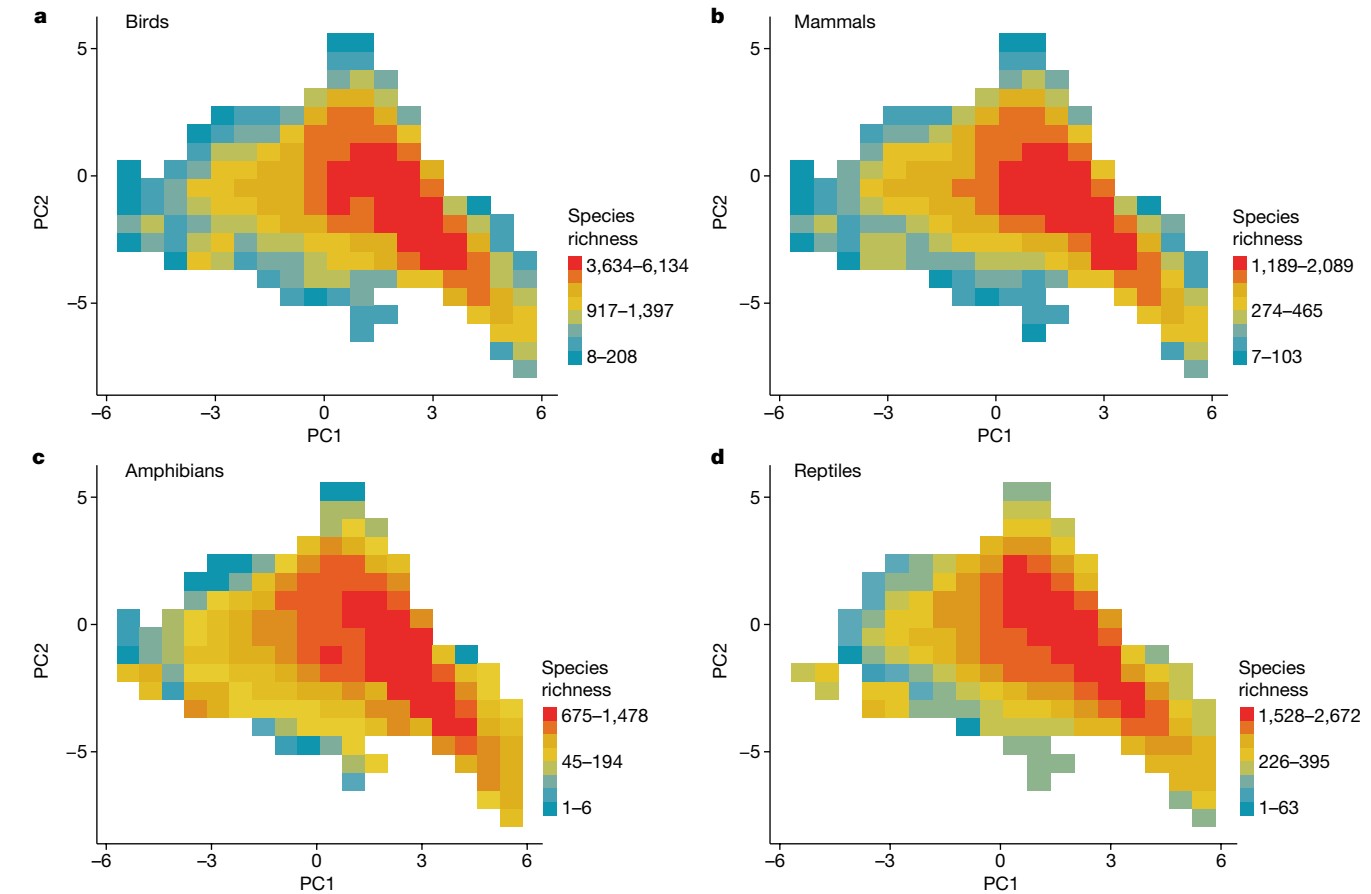

**Fig. 2 | Terrestrial tetrapod richness in climate space. a–d,** Number of species that fall within a climatic condition counted for birds ($n = 180$ climate cells; **a**), mammals ($n = 178$ climate cells; **b**), amphibians ($n = 174$ climate cells; **c**) and reptiles ($n = 165$ climate cells; **d**). PC1 and PC2 represent the first and second axes of a principal component analysis of 12 global-scale climate variables (Supplementary Information).

climate is structured globally and help face the challenges imposed by climate change.

We used the first two axes of a principal component analysis of 12 global-scale climate variables to define a two-dimensional orthogonal climate space (Fig. 1a–c) that represents thermal and water availability limits to species distribution. Each axis of the climate space was then divided into climate cells of equal climate intervals (Fig. 1c; results robust to interval size, Supplementary Information). Thus, each climate cell represents several geographic locations that fall within a specific climate interval. Likewise, several geographic locations belong to a unique climate cell (Fig. 1d). This connection between climate and geographic space, referred to as Hutchinson's duality[18], allows geographic information to be mapped in climate space and vice versa. Using this approach, we computed the geographic extent of a climatic condition that we refer to as climate area (Fig. 1f). As a climate condition is scattered on the surface of the planet, we identified the individual fragments of a climate condition (that is, regions within a climate cell that are geographically isolated from other regions within the same climate cell) and measured the average geodesic distance among climate fragments, which we refer to as climate isolation (Fig. 1e). Climate area and climate isolation represent the geography of global climate. We also computed the within-climate-cell average of the first two principal components for each climate cell to evaluate the effect of the climate itself. Finally, for each climate cell we counted the number of species that fall within that climatic condition (Fig. 2) using range distribution data for terrestrial amphibians, reptiles, mammals and birds, and investigated the effect of the climate, and its area and isolation on richness patterns and species composition of each tetrapod group.

For each tetrapod group, we fitted a Poisson-distributed generalized additive mode to account for nonlinearity in model residuals with species richness in climate space as the response variable, and the geography of climate (that is, climate area and isolation) and climate itself (that is, first and second principal components) as predictors (Extended Data Table 1). Our model explained nearly 90% of the variation in richness for birds (proportion of null deviance = 0.90, adjusted $R^2$ = 0.92), mammals (proportion of null deviance = 0.90, adjusted $R^2$ = 0.91), amphibians (proportion of null deviance = 0.88, adjusted $R^2$ = 0.90) and reptiles (proportion of null deviance = 0.88, adjusted $R^2$ = 0.91)– results hold consistent when nonlinearity is addressed through polynomial regressions (Supplementary Information). These results indicate that for all tetrapod groups, the geography of climate and climate itself can explain most of the variability of tetrapod richness in climate space. For all groups (Extended Data Figs. 1–3), partial residual richness (that is, richness not explained by other predictors) increases with area (Fig. 3a) and with isolation (Fig. 3b), even though climates occurring on opposite poles are isolated, their extreme conditions are suitable to only a few species. Thus, the geographic distribution of polar climates does not effectively isolate a substantial number of species. Partial residual richness is positively related to the first principal component (Fig. 3c) that is composed mostly by temperature variables and has a hump-shaped relationship with the second principal component that is defined by a balance of energy and water availability (Fig. 3d). The explanatory power of the geography of climate and climate itself is very similar for all groups (Fig. 4). In terms of proportion of null deviance, climate area contributes about 10%, whereas climate isolation accounts for roughly 5%. Comparatively,

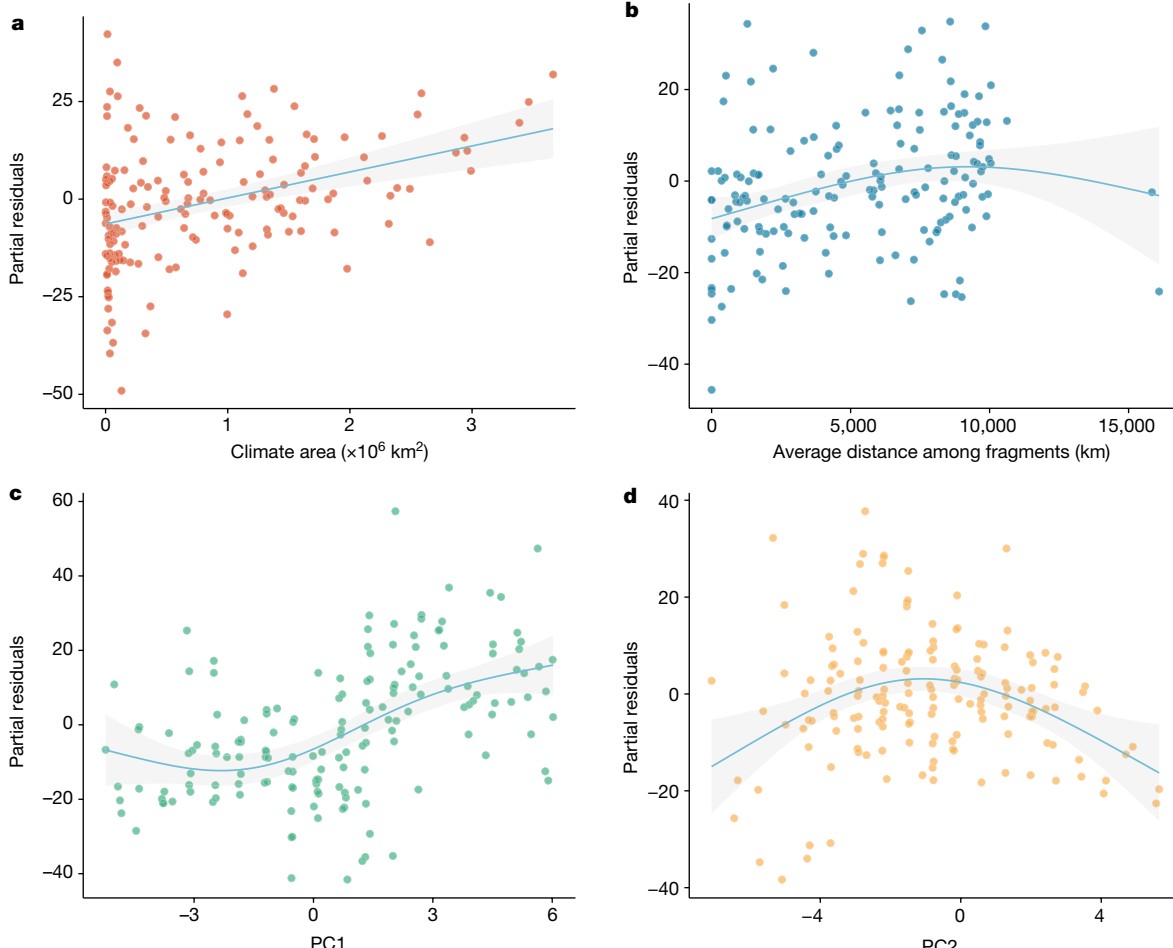

**Fig. 3 | Relationships between partial residual richness of birds (that is, richness not explained by other predictors of the multivariate model) and each model predictor. a,b,** Partial residual richness versus climate area and isolation. **c,d,** Partial residual richness versus climate itself here represented by the first and second principal components. The grey shading around each regression line represents the 95% confidence interval. Identical patterns are observed for other tetrapod groups (Extended Data Figs. 2 and 3).

the first and second principal components contribute around 13% and 2%, respectively. Shared explanatory power within the geography of climate totals about 12%, whereas within climate itself it is around 2%. Thus, when considering both isolated and shared contributions, the impact of the geography of climate is nearly double that of climate itself (Fig. 4). The remaining 45% of shared explanatory power comes from overlapping contributions of geography of climate and climate itself. Using different resolutions and combinations of variables to define climate space did not qualitatively change the results (Supplementary Information).

Our results reinforce previous findings that the geographic extent of climatic conditions is positively associated with species diversity[7,11,12], but further highlight that only 10% of the variation in richness comes exclusively from climate area alone. The mechanisms underlying the species–area relationship in climate space are different from those proposed for geographic space because similar climatic conditions are not continuously distributed across geography. In geographicl space, a continuous large area often leads to higher environmental heterogeneity and consequently different species exploring different ecological opportunities[24]. However, scattered climatic conditions in climate space, that sum up to define climate area, represent several geographic locations within the same type of environment. As a result, within a given climate, there is little environmental heterogeneity (assuming climate variation influences environmental heterogeneity) that could lead to specialization for different environmental conditions.

Therefore, the finding of more species occurring with homogeneous climatic conditions that occupy larger geographic extents is probably attributable to capacity rules[25]. Climatic conditions that cover more extensive land areas are believed to support more individuals, leading to larger populations. Larger populations, in turn, are associated with increased rates of speciation and reduced rates of extinction[5,25,26].

Another aspect of the geography of climatic conditions is that a larger climate area does not necessarily translate to larger continuous habitat. In fact, climate area is strongly correlated with the number of climate fragments (that is, geographic cells connected to each other that occur within a single climate; Pearson's $r = 0.95$; Fig. 5a and Extended Data Fig. 4). However, the correlation between the number of climate fragments or climate area and climate isolation (measured as the average distance between climate fragments) is weak, with Pearson's correlation coefficients of 0.23 and 0.11, respectively. These observations highlight the importance of characterizing climate isolation in addition to the area of climatic conditions when macroecological patterns are analysed in climate space.

The mechanism underlying the association between species richness and climate isolation is probably linked to climate isolation influencing gene flow among diverging populations. Over deep time, at the global scale, climate change, continental drift and mountain uplift can affect the spatial connection among similar climatic conditions. Populations of a species dispersing to follow their optimum climatic conditions expand, contract and fragment their geographic distribution within

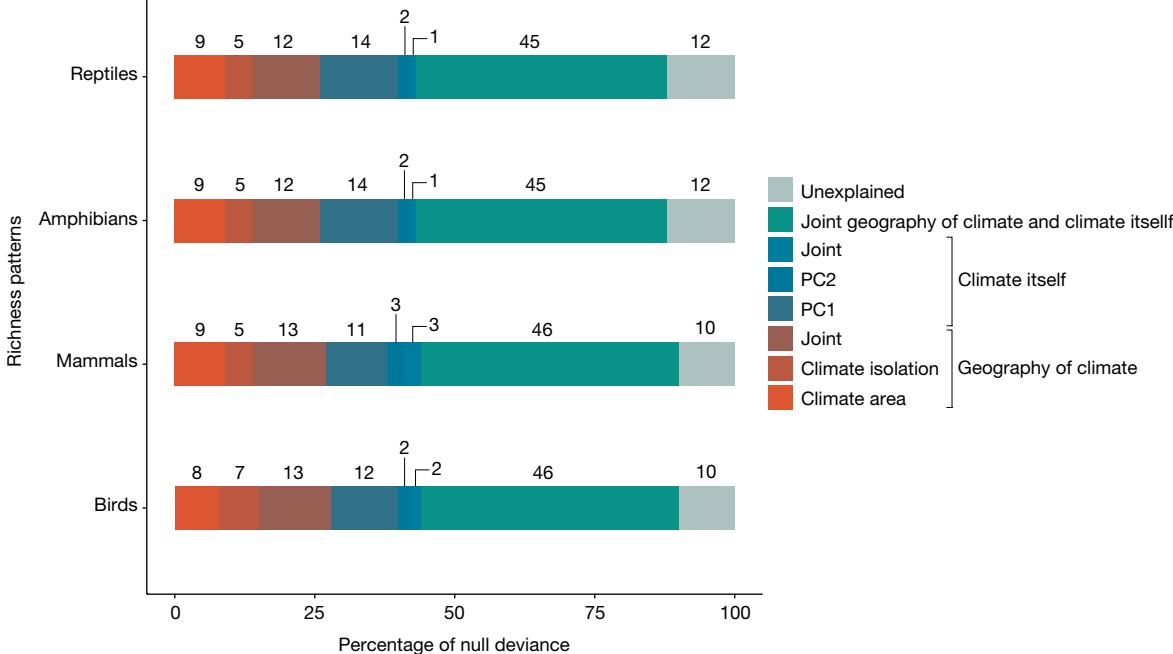

**Fig. 4 | Relative contribution of climate and the geography of climate.** For each tetrapod group, we separated how much of the variability in species richness on the climate space is explained exclusively by climate area and climate isolation (geography of climate) and principal components one and two (climate itself). We also computed the joint contribution within the geography of climate and climate itself and between the geography of climate and climate itself.

climatic conditions[27]. The isolation of populations within spatially disconnected climatic conditions increases the chances of allopatric speciation events and founder speciation events caused by long-distance dispersal[27,28]. Therefore, at the global scale we expect that locations where climatic conditions are more isolated will also have more species, which is consistent with our findings. In addition, isolated climatic conditions shelter biotas that independently evolved within that climatic condition making it likely that different species pools will be sampled. This is in fact evidenced by the increase in the turnover of species composition with the increase of climate isolation (Fig. 5c).

Notably, the most conspicuous and discussed pattern of climate isolation at the global scale is that of two broad climatic zones in high latitudes, with northern and southern polar and temperate zones being disjunct and separated by a large contiguous tropical zone[8,29]. However, our results show that tropical climates, spread across several continents, are more isolated than polar and temperate climates. There is an envelope relationship between the distance of similar climates (distance among climate fragments) and latitude (Fig. 5b and Extended Data Fig. 5), with a tendency of shorter distances among similar climates at higher latitudes (Fig. 5b, ordinary least squares model) and greater variability in geographic isolation of similar climates in tropical zones. The general tendency of greater isolation in lower latitudes increases the probability of independent pools of species evolving within warmer and more humid climates spread in a mosaic of similar but isolated climatic conditions. We find that climates occupying large and isolated areas have increased community differentiation with both area and isolation of climate (Fig. 5c,d and Extended Data Figs. 6–8), indicating that ecological communities occurring within larger and isolated climates have high replacement of species among communities. The likely mechanism behind this pattern is that independent pools of species evolve as a consequence of dispersal limitation and historical changes in the geography of climate[30]. In addition, climates occupying small and connected areas have a nested community structure with fewer species that are a subset of richer communities within climates (Fig. 5e,f and Extended Data Figs. 6–8). This pattern probably emerges

because of lower dispersal limitation within smaller and connected climatic conditions. These results demonstrate how the area and isolation of climate strongly capture changes in community composition even within homogeneous climatic conditions. Thus, climate area and isolation capture patterns of community differentiation across the globe.

The area hypothesis to explain latitudinal diversity gradients emerged through observations that climatic conditions that occur in tropical environments are more common than climatic conditions that occur in extratropical regions[7,8]. However, with the observed reduction in climate isolation at higher latitudes, it becomes evident that not only the commonness of climatic conditions, but also the geographic distribution and isolation of similar climatic conditions, need to be taken into account. Here we propose that the area hypothesis to explain global-scale patterns of species diversity should be modified into an area–isolation hypothesis because not only do lower-latitude climates have larger geographic extent, but climate isolation decreases towards the poles. Thus, tropical climates, characterized by large areas that are both fragmented and isolated, tend to have more observed species. Such climatic structures could promote greater speciation rates through capacity rules and reduced gene flow. Our results show: (1) increase of species richness and turnover with climate area and isolation; (2) larger isolation of tropical climates; and (3) the degree of fragmentation of larger climates call for a revision of the area hypothesis. Although climates occurring in the tropics are more common, these climates are also more fragmented and isolated.

Even though our model, considering all geographic features of climate, explains a large fraction of the variation in tetrapod diversity, the remaining 10% of unexplained variation showed interesting patterns when projected to climate and geographic space (Fig. 6). Along with the model's goodness of fit, residual patterns are consistent among tetrapod groups (Fig. 6). These residual patterns highlight an important aspect of analysing species diversity directly in climate space. Historical contingencies of different regions with the same climatic conditions are disregarded by combining the presence and absence of all species in those regions, regardless of whether the regions are clustered or

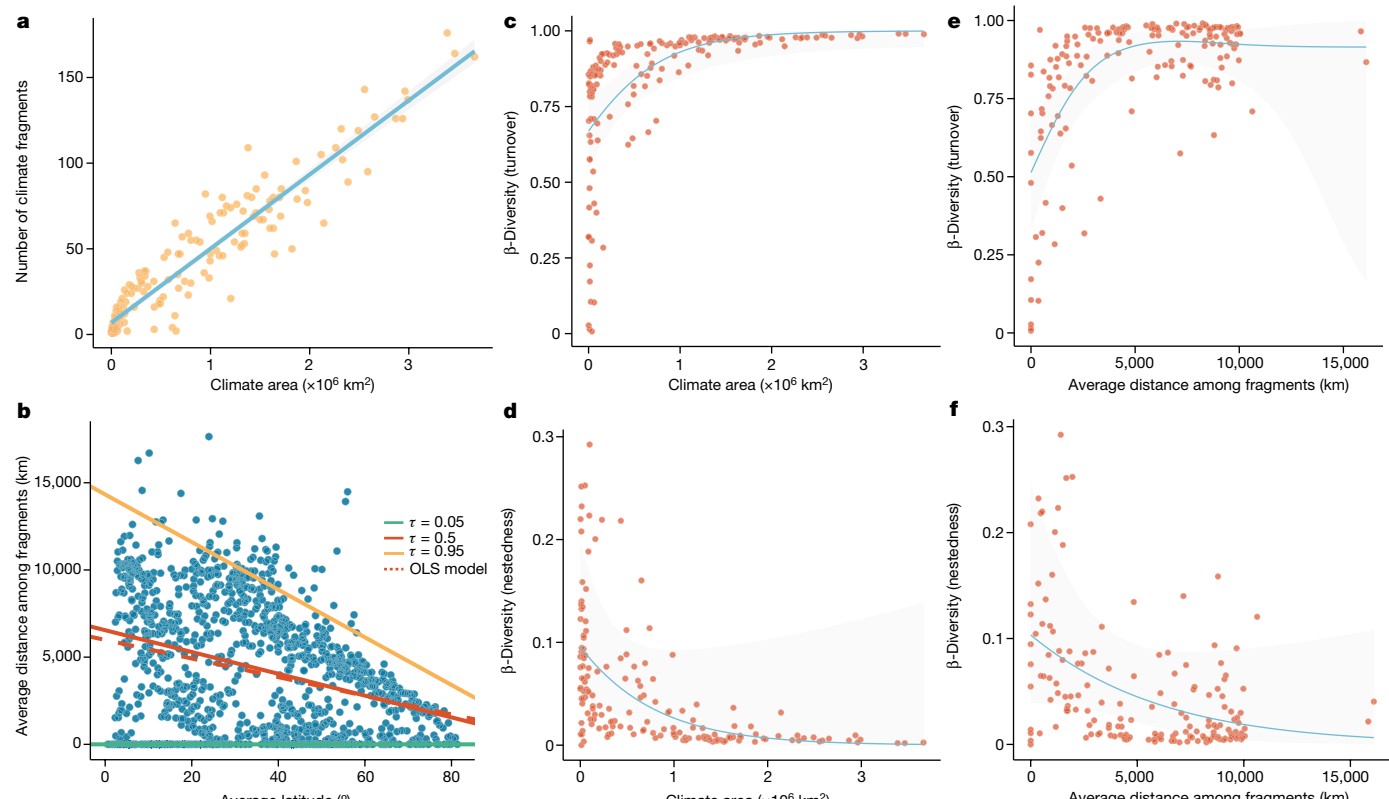

**Fig. 5 | The geography of climate across the globe and its relationship with bird community composition. a**, The relationship between climate area and the number of climate fragments (*n* = 164 climate cells; Extended Data Fig. 4). **b**, The relationship between average absolute latitudes of each climate cell in climate space and climate isolation (*n* = 1,028 climate cells) at a resolution of 60 equal intervals (Extended Data Fig. 5). In **b**, *τ* expresses the quantiles in which quantile regressions are fitted and OLS represents the ordinary least squares regression (dashed line). **c,e**, The relationship between the geography of climate and the turnover component of taxonomical β-diversity (*n* = 164 climate cells; Extended Data Figs. 6–8). **d,f**, The relationship between the geography of climate and the nestedness component of β-diversity (*n* = 164 climate cells; Extended Data Figs. 6–8). The grey shading around regression lines represents the 95% confidence interval.

scattered around the globe. However, geographic patterns of model residuals can give insights into why more or fewer species are observed under certain climatic conditions after controlling for the effects of the geography of climate and climate itself.

With few exceptions, parts of mountainous climates across all continents have more species than expected by the climate and climate geography. However, the spatial resolution in our global-scale study may not be enough to precisely characterize mountainous climates. These climates usually occupy a small geographic extent (25% of terrestrial landmass[31]) when compared to lowlands and yet, especially in tropical environments, are home to approximately 90% of tetrapod species[31]. Tropical mountains usually have more species than expected by climate itself (that is, positive residuals[31,32]) and the same occurs when the geography of climate is also considered (Fig. 6). Using the present climate to define the effect of climate on mountainous regions misses one important aspect, namely the great importance of mountains for biodiversity refugia under climatic oscillations[31]. Therefore, both the area and isolation of climate and climate per se are not enough to explain the building of mountain diversity, especially in tropical mountains such as the Andes.

Lower species richness than expected by climate itself and the geography of climate (that is, negative residuals) are observed in many arid regions across the world with few exceptions. In South America, the eastern dry diagonal connecting Caatinga, Cerrado and Chaco and the western dry diagonal connecting Patagonia, Monte, Prepruna dry Pruna and the Atacama Desert, and in Africa, part of the Sahara Desert and the Somalian Desert, among other dry regions (Fig. 6), show a very clear pattern of fewer species than expected by the geography of climate and climate itself. Contrary to mountainous regions, dry regions occupy a large geographic extent, which given the expected species–area relationship, should support a greater number of species. However, their extreme climates act as a barrier for lineages that are not capable of surviving in such arid environments[33]. The difficulty of adapting to extreme conditions[33], even if these conditions are common on the surface of the planet, could explain why fewer species are observed in these regions than expected by climate itself and the geography of climate.

Here we demonstrated that the present geography of climate and climate itself can explain a large fraction of the tetrapod diversity and that the isolated effect of the geography of climate almost doubles the effect of climate itself. However, species richness might not be associated only with the present climate, but also with past climate and past geographic structures of climate. It should be expected that over deep time some climatic conditions were more common than others, that connections and disconnections of climatic conditions occurred and that some conditions appear and disappear across millions of years. Therefore, exploring the dynamics of the geography of climate over deep time is a natural next step for studying diversity–climate relationships in climate space. The same rationale for past climatic dynamics can also be used for future climate change. Ongoing climate changes may alter the commonness of climatic conditions as well as their connection and isolation. For example, climate velocity, representing the direction and speed that species move to maintain their current climatic condition under climate change[34,35], is largely affected by climate connectivity[36] and can benefit from better understanding of the effect of the geography of climate on biodiversity patterns. If many species

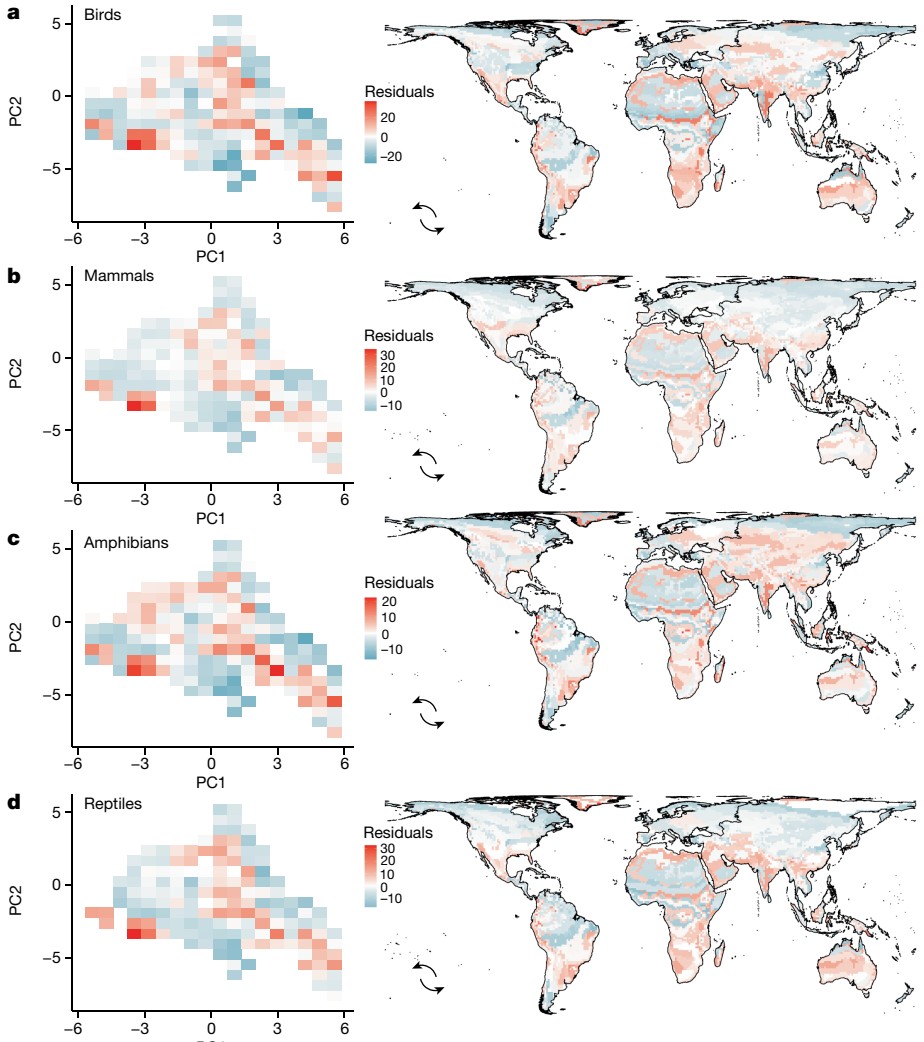

**Fig. 6 | Species richness that is not explained by climate itself and the geography of climate (that is, model residuals). a–d**, The residual richness is mapped in climate and geographic space for birds (**a**), mammals (**b**), amphibians (**c**) and reptiles (**d**). Blue areas express locations where fewer species are predicted than expected by model predictors and red areas express locations where more species are predicted than expected by model predictors.

have their optimum climate within a climatic condition that contracts its geographic extent with climate change, then competition might increase, potentially influencing species coexistence. In addition, the contraction of the geographic extent of a climatic condition imposes evolutionary pressure for individuals to shift their optimum to other similar climatic conditions that are either less saturated with species or have expanded their geographic extent on the surface of the planet. Finally, connections and disconnections of climate affect the ability of species to disperse between similar climatic conditions affecting the isolation among populations and consequently influencing their gene flow. Therefore, understanding how the geography of climate is associated with biodiversity is key to better understand and mitigate the impacts of climate change on biodiversity.

Building on these insights, we underline the urgent necessity to incorporate the geography of climate into studies examining the impact of climate change on biodiversity patterns. This crucial element has been largely disregarded. Recognizing the demonstrated connection between these geographic factors and species richness, it is critical that we unravel their temporal dynamics for effective biodiversity conservation. We must look beyond mere changes in climate and also consider their commonness and isolation across the planet. Neglecting these factors may result in unexpected climate change impacts

on biodiversity, stemming from our failure to track changes in the geography of climate.

Key considerations that emerge include whether our current conservation efforts are inadvertently biased towards protecting common climates, thereby neglecting rarer climates that may harbour unique species. Furthermore, we must understand how ongoing climate change might influence the prevalence and isolation of various climatic conditions. Climate change could fragment previously continuous climates, intensifying the challenges for species to disperse and maintain their preferred climatic conditions. Likewise, if a previously extensive climate shrinks owing to climate change, this could influence species coexistence.

In essence, to mitigate the impacts of climate change on biodiversity, it is paramount that we deepen our understanding of the geography of climate and its shifts over time. This approach could represent a substantial contribution to conservation biology, providing more comprehensive and effective strategies for biodiversity preservation.

## Conclusions

Here we show that the observation of greater species richness in warmer and more humid regions relies not only on climate itself but also on

how climate is distributed on the surface of the planet. We demonstrate that the geography of climate plays a key role in the diversity–climate relationship of tetrapods. Its effect can be separated from the effect of climate itself, and the amount of explanation attributed to the geography of climate is almost double the effect of climate itself. Unexplained variation can be clearly linked to historical processes among different regions that are removed when diversity is analysed directly in climatic dimensions. Moreover, shifting the focus of macroecological studies from geographic to climate space allowed us to describe basic properties of the geography of climate. We showed that climate area can precisely capture the number of climate fragments across the globe but is not a good representative of climate isolation. Finally, we showed that climate isolation decreases towards the poles, but has a high variability in the tropics.

Our model, considering both climate and climate geography, explains nearly all of the variability in species richness for each tetrapod group. Even though ectotherms do not occupy all available climates on the planet, ectotherms and endotherms are consistent in their response to climate and the geography of climate with similar patterns even when unexplained richness is mapped. Given the positive effects of climate area and isolation on species richness and how these geographic properties of the climate are structured on the surface of the Earth, we propose that the area hypothesis to explain large-scale patterns of species diversity should be merged into an area and isolation hypothesis. Such a shift in studies exploring diversity patterns directly in climate dimensions could bring a much-needed focus to how ongoing climate change can alter both the geographic extent and isolation of climatic conditions. Describing patterns at large taxonomic, spatial and temporal scales directly in climate space promises substantial new insights into macroecology and biogeography by revealing previously hidden effects of climate on global biodiversity patterns.

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

## Methods

### Biological data in geographic space

We obtained vector range maps of all amphibians and mammals available from the International Union for Conservation of Nature (https://iucn.org). We obtained bird and updated squamate range maps, respectively, from BirdLife (version 2020.1; http://datazone.birdlife.org) and ref. 37 (Global Assessment of Reptile Distributions internal version 1.5). Resident and native current distribution ranges of the species were gridded at about 110-km resolution with Behrmann equal area projection (approximately 1° resolution). This resolution has been suggested as the most appropriate grain for this type of data[38,39], being less prone to incur false presence of species at the global scale, our geographic scale of exploration in this study.

### From geographic to climate space

Species distributions and their habitats can inform us about the *n*-dimensional hypervolume of biotic and abiotic conditions in which each species can maintain viable populations[17,18]. As the climatic information underlying a species' distribution (that is, Grinellian niche) is easier to access than biotic conditions (that is, Eltonian niche), an *n*-dimensional environmental space, built with climatic variables (that is, climate space), can be constructed and used to measure biodiversity within climate space. Here we defined a two-dimensional climate space that considers the limits imposed by temperature and water availability for terrestrial species. A two-dimensional climate space allows us to use the same methodologies to study diversity patterns as traditionally used in two-dimensional (that is, longitude and latitude) geographic space. To establish each axis, we first averaged several current climatic variables within each spatial grid cell of approximately 110-km resolution. These variables describe variations in temperature and water availability, and include mean annual air temperature, mean diurnal air temperature, isothermality, temperature seasonality, mean daily maximum air temperature of the warmest month, mean daily minimum air temperature of the coldest month, annual range of air temperature, annual precipitation amount, precipitation amount of the wettest month, precipitation amount of the driest month, precipitation seasonality and potential evapotranspiration. Subsequently, we computed the principal components on the correlation matrix of these averaged variables (Supplementary Information). We obtained the bioclimatic variables from the Climatologies at High Resolution for the Earth's Land Surface Areas (CHELSA; https://chelsa-climate.org/bioclim/) dataset[40] and potential evapotranspiration from the CGIAR (Consultative Group on International Agricultural Research) Consortium for Spatial Information (https://cgiarcsi.community)[41]. The two axes of the principal component analysis captured about 80% of the global variation in climate. Using different combinations of variables, such as replacing potential evapotranspiration by net primary production, also obtained from CHELSA[40] (Supplementary Tables 3–6), log transforming skewed variables or defining a two-dimensional climate space with mean annual temperature and mean annual precipitation (Supplementary Tables 7–10) did not qualitatively change the results (Supplementary Information). Defining a space with independent axes can be complex when dealing with correlated variables, such as mean annual temperature and precipitation. Their interrelation can challenge the concept of a genuine two-dimensional space, similar to geographic mapping with longitude and latitude. To overcome this challenge, we used principal components from a principal component analysis to define our climate space in the primary results. This method not only incorporates several variables reflecting energy and water availability constraints but also ensures the creation of truly orthogonal climate axes. Future studies of different organism groups should define an environmental space in line with the specifics of the group under investigation, as different organisms (for instance, aquatic versus terrestrial) might face different limiting variables.

We extracted the scores of each principal component for the gridded map at about 110 km (Fig. 1a,b) that were used to define species presence and absence for tetrapod groups. Choosing a procedure similar to that of defining a gridded geographic map through intervals of latitudinal and longitudinal degrees (or kilometres), we defined a gridded climate domain by assuming intervals of equal size within the climatic axes. Based on adjustable equal intervals across the climatic axes, a gridded climate domain can be defined at any desired resolution. Resolutions below 20 intervals in each climate variable produce a small number of climate cells that are too few to fit the nonlinear models used in our study. Resolutions above 60 intervals in climate variables do not substantially increase the amount of climate cells. Therefore, we tested the sensitivity of our model to different resolutions of climate space by defining climate spaces with 20, 30, 40, 50 and 60 equal intervals attributed to each climatic variable (Supplementary Information). We showed the results for 20 equal intervals in the main text, which is the resolution that maximizes model fit for all groups. Results are consistent for all resolutions and tetrapod groups (Supplementary Information). On the one hand, each climate cell of the climate domain represents several geographic locations that fall within the climatic intervals defined by the two climatic variables (Fig. 1). On the other hand, each geographic location belongs to a unique climate cell (Fig. 1). These properties represent the duality between geographic and climate space[18], allowing geographic information to be mapped in climate space and vice versa.

The presence and absence information of species within climate cells is thus dependent on the geographic locations that occur within a given climatic interval. With this information, species richness is computed for all tetrapod groups by summing the presence of species within climate cells. The presence and absence information of species on the geographic cells that occur within a given climatic interval is also used to compute the multi-site β-diversity within a climate cell. The turnover component of β-diversity is computed through the Simpson index of dissimilarity, and the nestedness component of β-diversity is computed through the difference between Simpson and Sorensen dissimilarities[42]. The area of each climate cell (that is, climate area) is computed by summing the land surface area of all the geographic cells that occur within each climate cell (that is, all geographic cells with that respective climatic condition). To better represent climate isolation, we used three alternative metrics, but because some of them are strongly correlated with climate area and also correlated to each other we chose one of the three for the final analysis. We measured climate isolation as (1) the mean geodesic distance among all geographic cells that occur within a climate cell. Alternatively, we considered that some geographic cells that occur within a climate cell form fragments of climates (geographic cells connected to each other that occur within a single climate). Therefore, the size of climate fragments could affect the isolation measures. To control for this effect, we measured (2) the average distance between climate fragments that occur within a climate cell and (3) the number of climate fragments within a climate cell. The total count of fragments within a climate cell has a strong link with the overall climate area (with a Pearson's correlation of 0.94). Similarly, the average distance between all geographic cells within a climate cell is also strongly related to the average distance between climate fragments (Pearson's correlation of 0.88). In a process in which we calculate the average distance between all geographic cells within a climate cell, larger climate fragments may have a greater impact on the average because they provide more points of comparison. Therefore, in our final models, we chose to use the average distance between climate fragments within a climate cell. Finally, to represent the effect of the climate itself we computed the average principal components (PC1 and PC2) for each climate cell.

### Statistical analysis

The main questions we investigated in our study are whether gradients of species richness (response variable) are associated with climate

(PC1 and PC2) and the geography of climate (climate area and isolation). Model predictors were also inspected for multicollinearity with the variance inflation factor but showed very low collinearity (VIF < 1.5).

As our response variable is richness, a count data value represented by an integer variable equal to or greater than 1, a Poisson link is a natural choice. We fitted a Poisson generalized linear model with richness as response variable and geography of climate (that is, area and isolation) and climate itself (PC1 and PC2) as predictors. The generalized linear model explains approximately 75% of richness as measured by the proportion of null deviance and $R^2$ (MacFadden's $R^2$; Supplementary Table 19); however, the linear model violates the assumptions of linearity, deeming estimated parameters uninterpretable (Supplementary Fig 15). Nonlinear models implemented either by polynomial generalized linear models and general additive models are equivalent (Supplementary Information). In the main text, we opted to use general additive models because these are extensions of generalized linear models, being a safer and simpler option over polynomial regressions, which were originally designed for linear regression with normally distributed errors (Supplementary Information). Thus, we replace the $\beta$-coefficients from linear regression with flexible functions that allow nonlinear association between the variables in our dataset. For this matter, we used Poisson generalized additive models with penalized smooth functions conducted through generalized cross-validation[43,44]. Generalized cross-validation trades-off curve complexity against goodness of fit to avoid complex overfitted estimates. Basis dimension choices for smooth terms were set to four ($k = 4$) as patterns in our residuals are not better explained by higher dimensions of $k$. In our general additive models assuming Poisson distribution, the response variable for each tetrapod group consisted of the richness observed in climate space and the predictors consisted of climate area, climate isolation and the average principal components of the first two axis of our climatic principal component analysis (PC1 and PC2). We assess the goodness of fit for our model with three metrics: proportion (or percentage) of null deviance, adjusted $R^2$ and predicted $R^2$. The percentage of null deviance, suitable for nonlinear models with non-normally distributed errors, measures the divergence between the observed data and the model's predictions. It compares the null deviance (the fit for a baseline model with only an intercept and, if applicable, an offset) to the residual deviance (the deviance of the fitted model). Lower deviance values indicate a better model fit. The adjusted $R^2$ is a modified version of a standard $R^2$ value that accounts for the number of predictors in the model, providing a more precise measure of the model's explanatory power. The predicted $R^2$ measures the association between the predicted values produced by the model and the observed data, thus reflecting the model's predictive capacity. Values across the three metrics were similar (Supplementary Information). We retain the percentage of null deviance in the main text as it is more suitable for nonlinear models with non-normally distributed errors.

On the basis of the complete model incorporating all predictors, we wanted to know how much of the proportion of null deviance can be attributed to each predictor alone, the joint effect of the geography of climate (climate area and isolation), the joint effect of climate itself (principal components one and two) and the joint effect between the geography of climate and climate itself. To do so, we fitted alternative models without each predictor and pairs of predictors and computed a reduction in deviance. In this case, the reduced models used the same smoothing parameters as the complete model.

Model residuals were inspected for climate autocorrelation using Moran's I correlogram[45] as the existence of climate autocorrelation can upward bias the significance of predictor variables and their coefficients. Spatial autocorrelation was removed by adding axes of principal components of neighbour matrices[45] that were created from a distance matrix among climate cells. As principal components of neighbour matrices are used only to check whether predictors remain significant in the absence of spatial autocorrelation in the model's residuals, the model's $R^2$ values are reported without the addition of axes of principal components of neighbour matrices because these spatially structured variables are added to our models to solve a statistic problem of upward bias in the significance (Extended Data Table 1). Finally, we used quantile regression to analyse the envelope pattern[46] emerging from the relationship between average absolute latitudes and climate isolation with lower and upper boundaries of the relationship modelled with 0.05 and 0.95 quantiles.

## Reporting summary

Further information on research design is available in the Nature Portfolio Reporting Summary linked to this article.

## Data availability

Global climate and biological data used in our study are open source. All climate data are available at CHELSA (https://chelsa-climate.org/bioclim/) and CGIAR (https://cgiarcsi.community). Vector range maps of all amphibians and mammals are available at the International Union for Conservation of Nature (https://iucn.org). Bird and squamate range maps are available, respectively, at BirdLife (version 2020.1, http://datazone.birdlife.org) and ref. 37 (https://doi.org/10.1038/s41559-017-0332-2). Source data are provided with this paper.

## Code availability

R code for statistical analyses and data tables are available as Supplementary Information.

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

**Acknowledgements** M.T.P.C. and C.H.G. are financially supported by the Swiss National Science Foundation (SNSF, no. 315230_197753). T.F.R. and J.A.F.D.-F. have been continuously supported by Conselho Nacional de Desenvolvimento Científico e Tecnológico (CNPq, grants PQ309550/2015-7 and PQ301799/2016-4) and by the National Institutes of Science and Technology (INCT) project in Ecology, Evolution and Biodiversity Conservation, supported by CNPq (grant 465610/2014-5) and FAPEG (grant 201810267000023). I.R.M. was supported by an SNSF Postdoc Mobility Fellowship (No. 206844). A.S., W.B. and L.P. were supported by the SNSF project Bigest (No. 310030_188550) and N.E.Z. was supported by the SNSF grant FeedBaCks (No. 20BD21_193907).

**Author contributions** M.T.P.C. conceived the study, developed the methods, handled all data processing, carried out analyses, generated the figures and led the writing with inputs from C.H.G., T.F.R., E.B., J.A.F.D.-F., N.E.Z. and D.W.R. All authors contributed to discussing and writing the study.

**Competing interests** The authors declare no competing interests.

**Additional information**
**Correspondence and requests for materials** should be addressed to Marco Túlio P. Coelho.

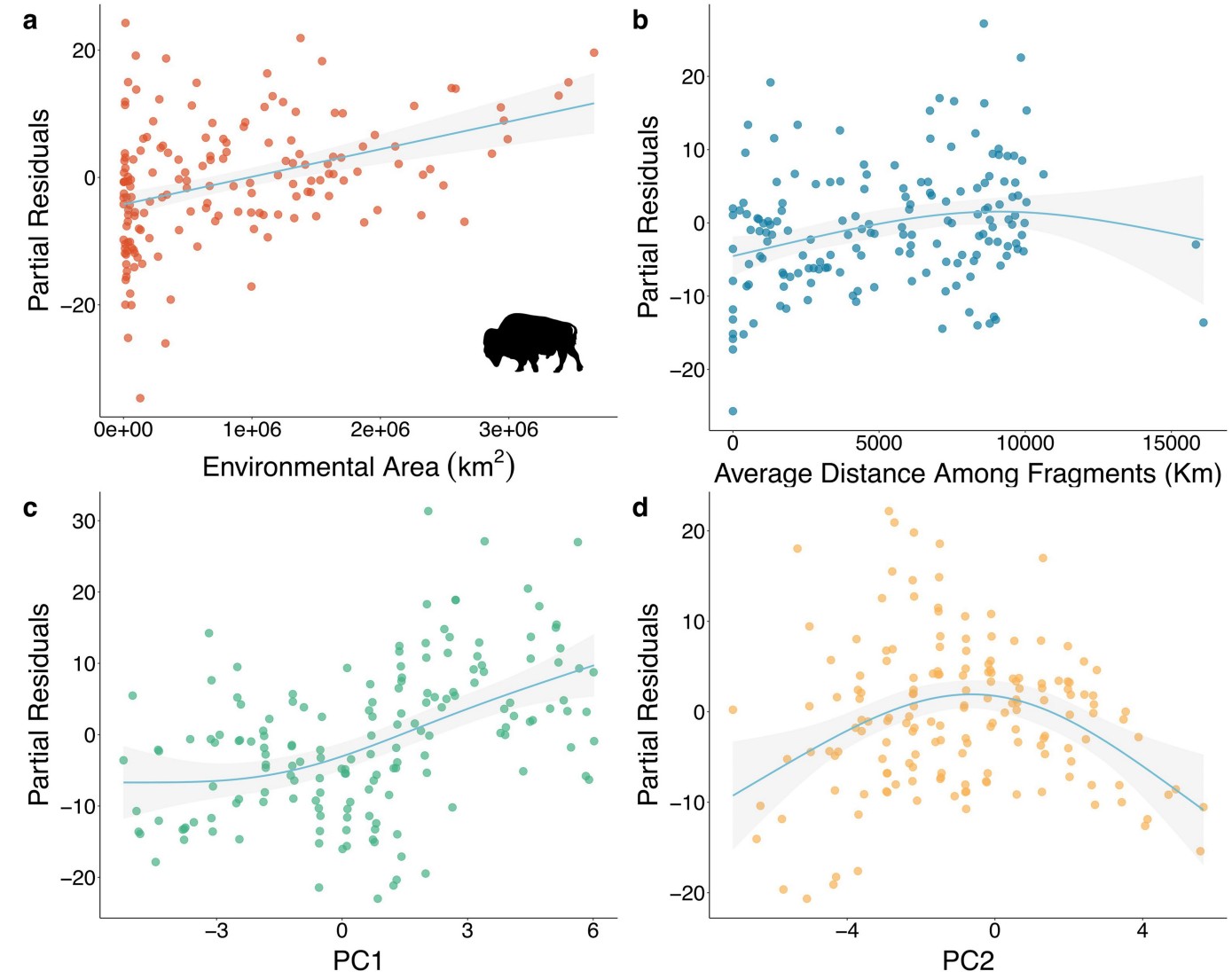

**Extended Data Fig. 1 | Relationships between partial residual richness (i.e., richness not explained by other predictors of the multivariate model) and all model predictors for mammals. a,b**, partial residual richness vs climate area. c,d, partial residual richness vs climate here represented by the first two principal components (n = 164 climate cells). The grey shading around each regression line represents the 95% confidence interval.

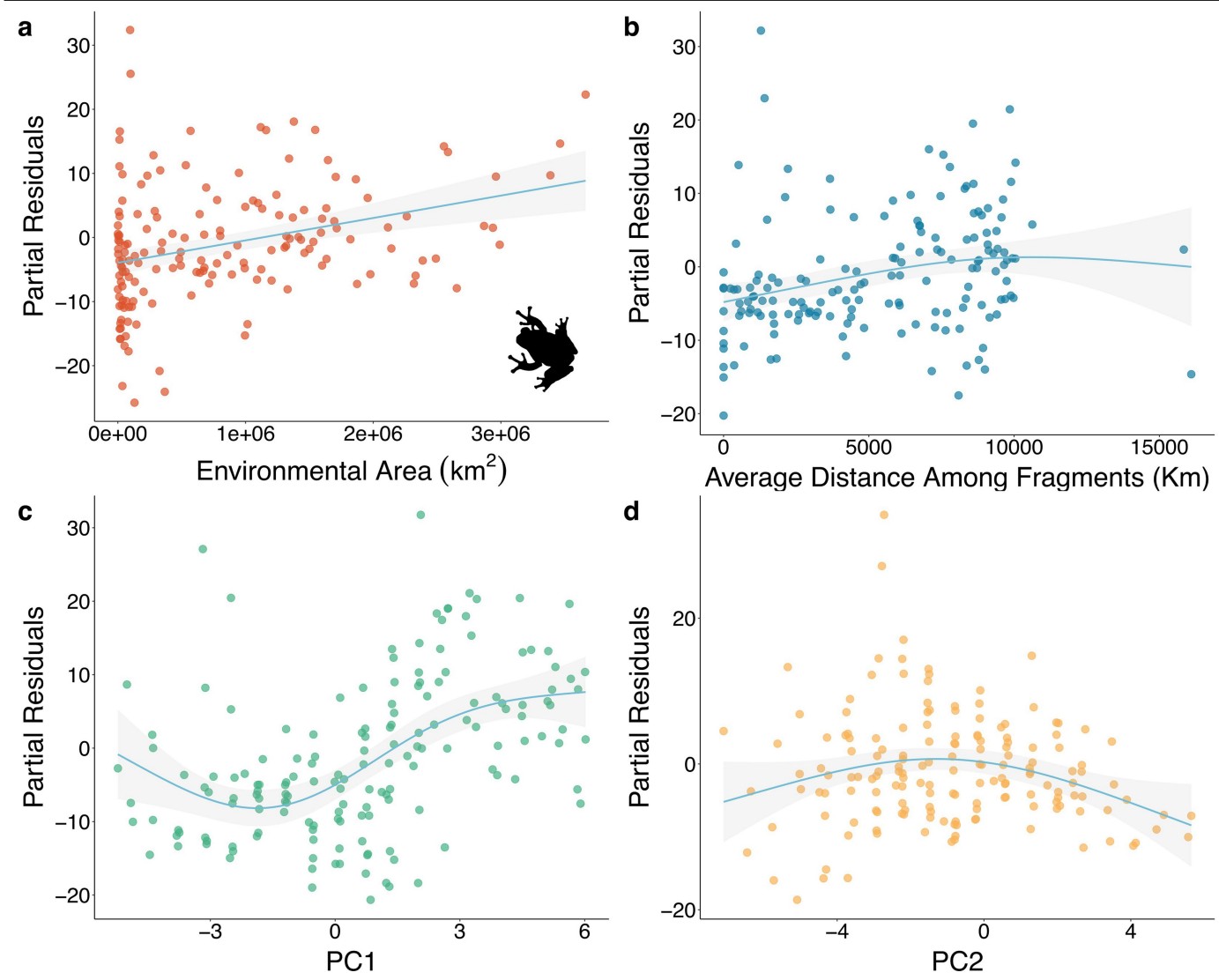

**Extended Data Fig. 2 | Relationships between partial residual richness (i.e., richness not explained by other predictors of the multivariate model) and all model predictors for amphibians. a,b**, partial residual richness vs climate area. **c,d**, partial residual richness vs climate here represented by the first two principal components (n = 159 climate cells). The grey shading around each regression line represents the 95% confidence interval.

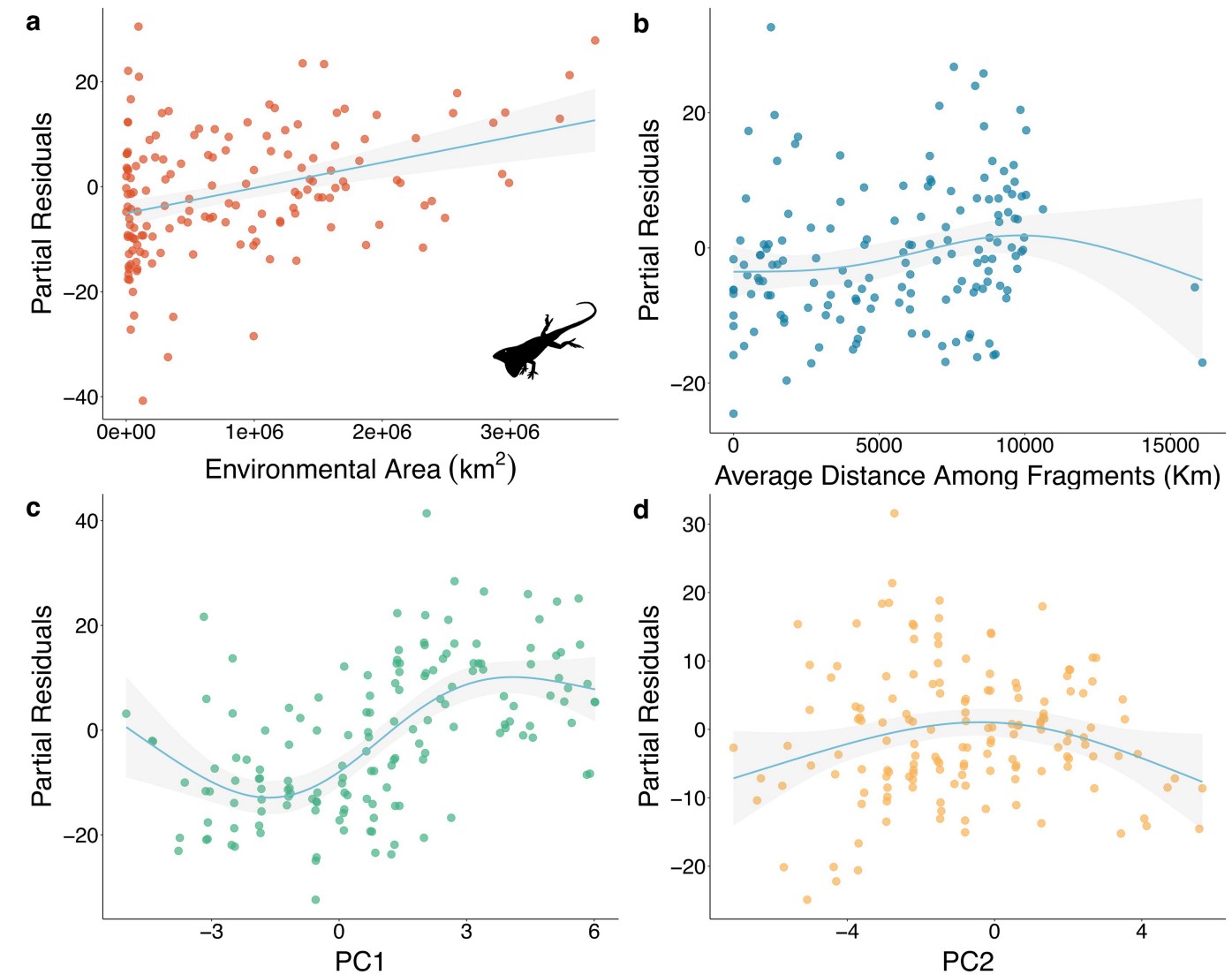

**Extended Data Fig. 3 | Relationships between partial residual richness (i.e., richness not explained by other predictors of the multivariate model) and all model predictors for reptiles. a,b**, partial residual richness vs climate area. **c,d**, partial residual richness vs climate here represented by the first two principal components (n = 151 climate cells). The grey shading around each regression line represents the 95% confidence interval.

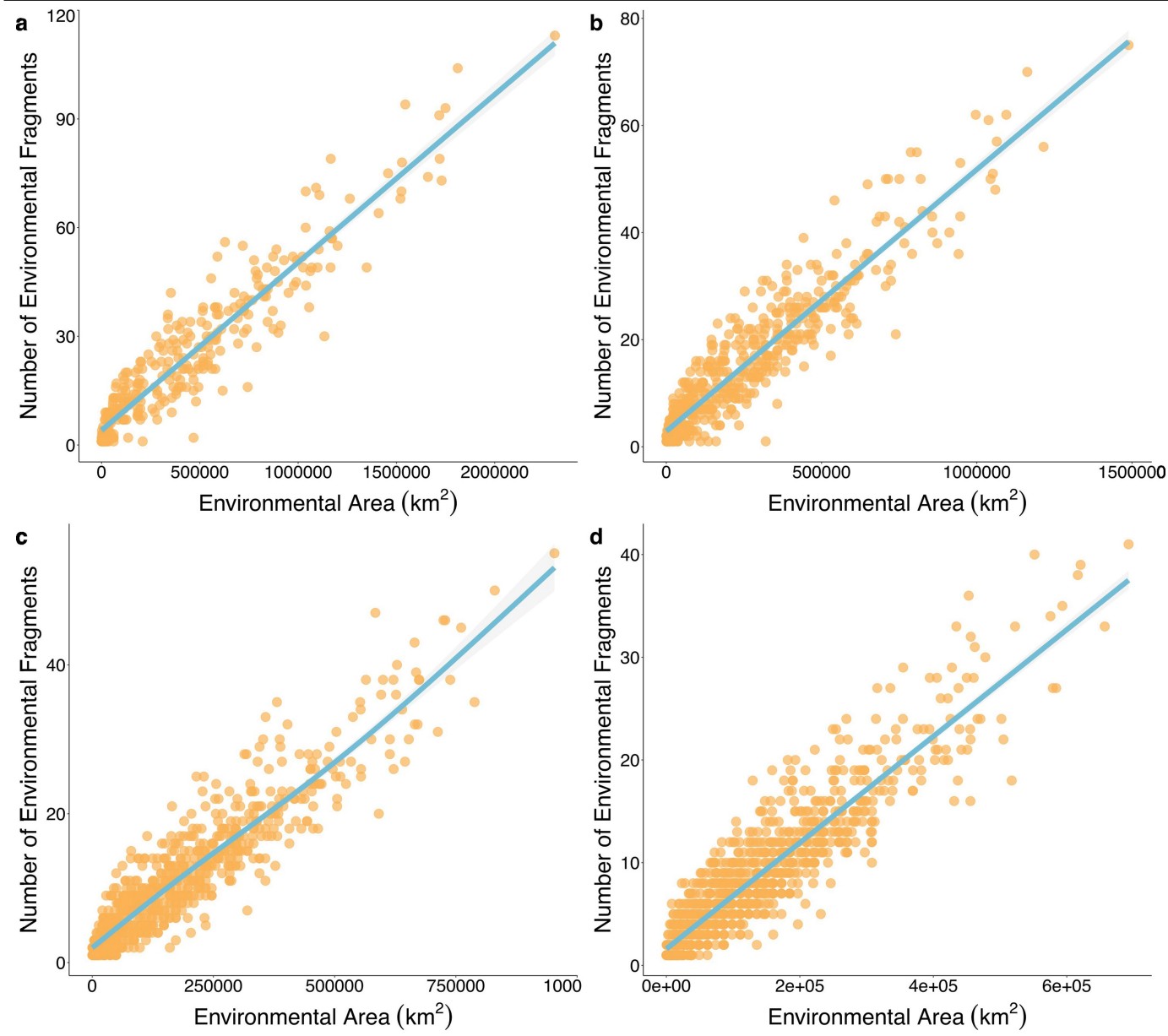

**Extended Data Fig. 4 | The relationship between climate area and the number of climate fragments across different resolutions.** From a to d, gridded climate spaces were built based on 30 (n = 326 climate cells), 40 (n = 525 climate cells), 50 (n = 756 climate cells) and 60 (n = 1028 climate cells) equal interval divisions of the climate axis. The grey shading around each regression line represents the 95% confidence interval.

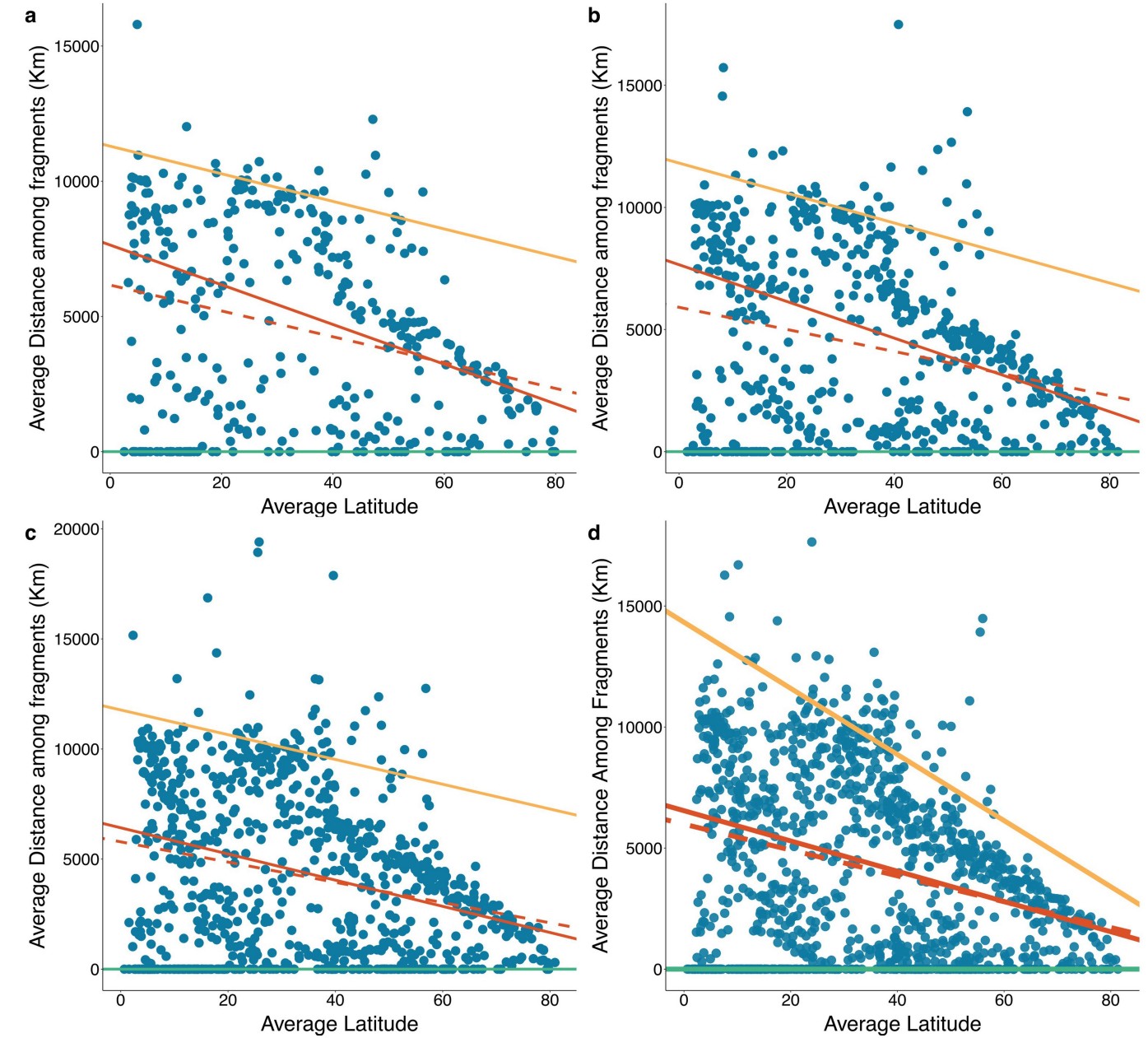

**Extended Data Fig. 5 | The relationship between average absolute latitudes of each climate cell in climate space and climate isolation across different resolutions of climate space.** The green line at the bottom represents a quantile regression where τ = 0.05. The yellow line at the bottom represents a quantile regression where τ = 0.95. The continuous red line represents a quantile regression where τ = 0.5. The dashed red line represents the ordinary least square regression. Here, the same relationship is shown for 30 (n = 326 climate cells), 40 (n = 525 climate cells), 50 (n = 756 climate cells) and 60 (n = 1028 climate cells) equal divisions of the climatic axis when defining climate space.

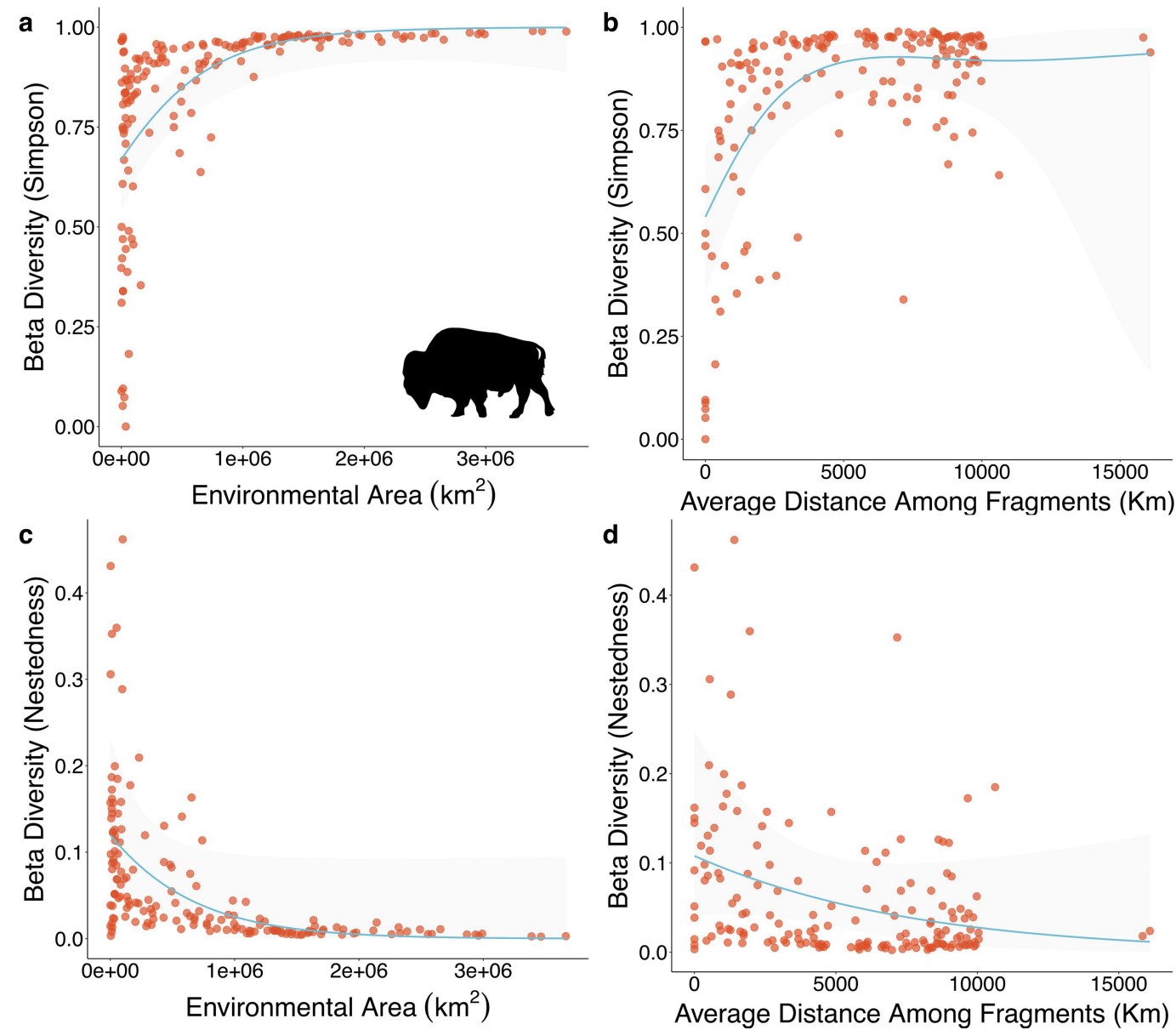

**Extended Data Fig. 6 | The geography of climate and its relationship with community composition for mammals. a**, **b** the relationship between the geography of climate and the turnover component of beta-diversity. **c**, **d** the relationship between the geography of climate and the nestedness component of beta-diversity (n = 164 climate cells). The grey shading around each regression line represents the 95% confidence interval.

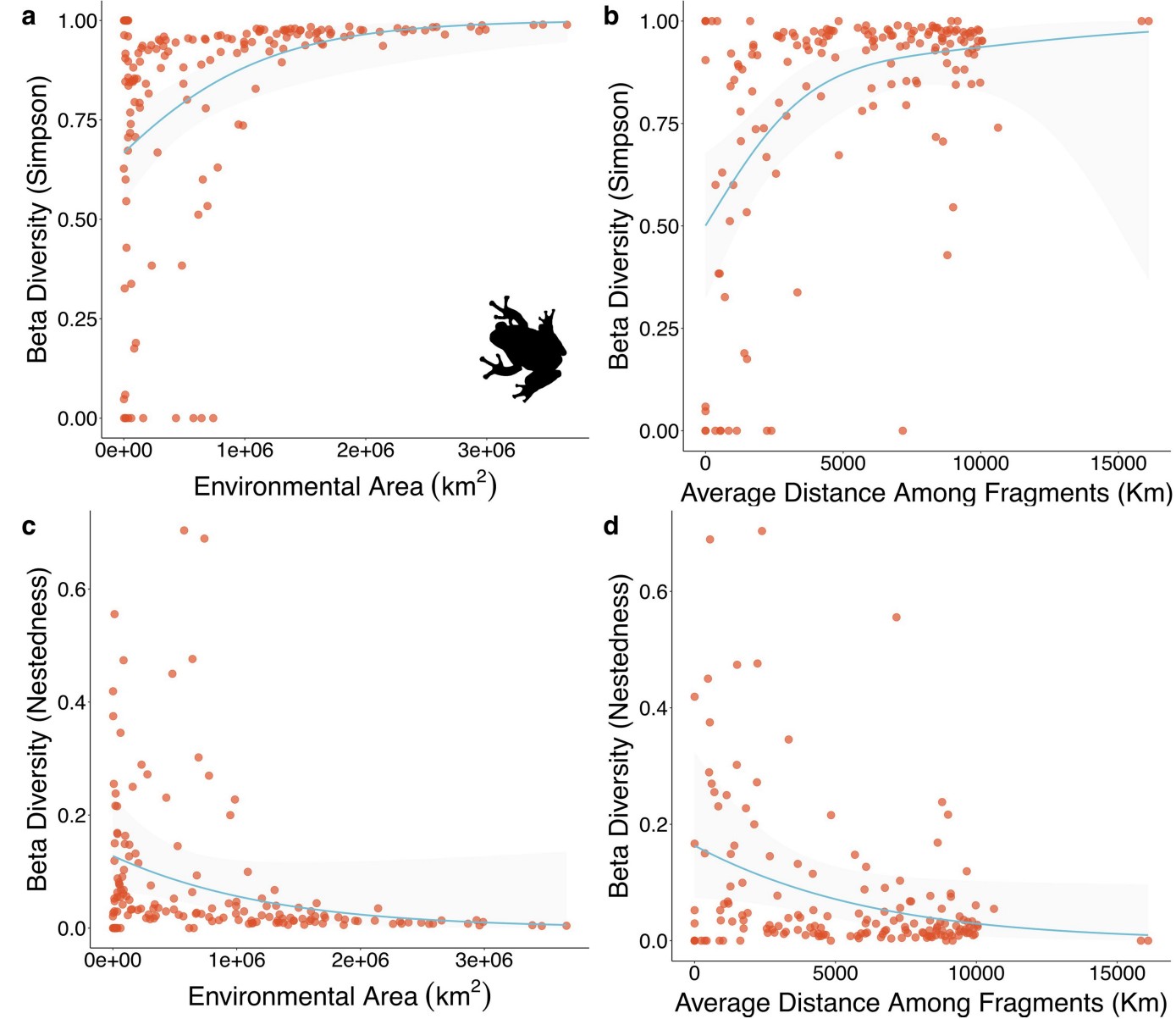

**Extended Data Fig. 7 | The geography of climate and its relationship with community composition for amphibians. a**, **b** the relationship between the geography of climate and the turnover component of beta-diversity. **c**, **d** the relationship between the geography of climate and the nestedness component of beta-diversity (n = 159 climate cells). The grey shading around each regression line represents the 95% confidence interval.

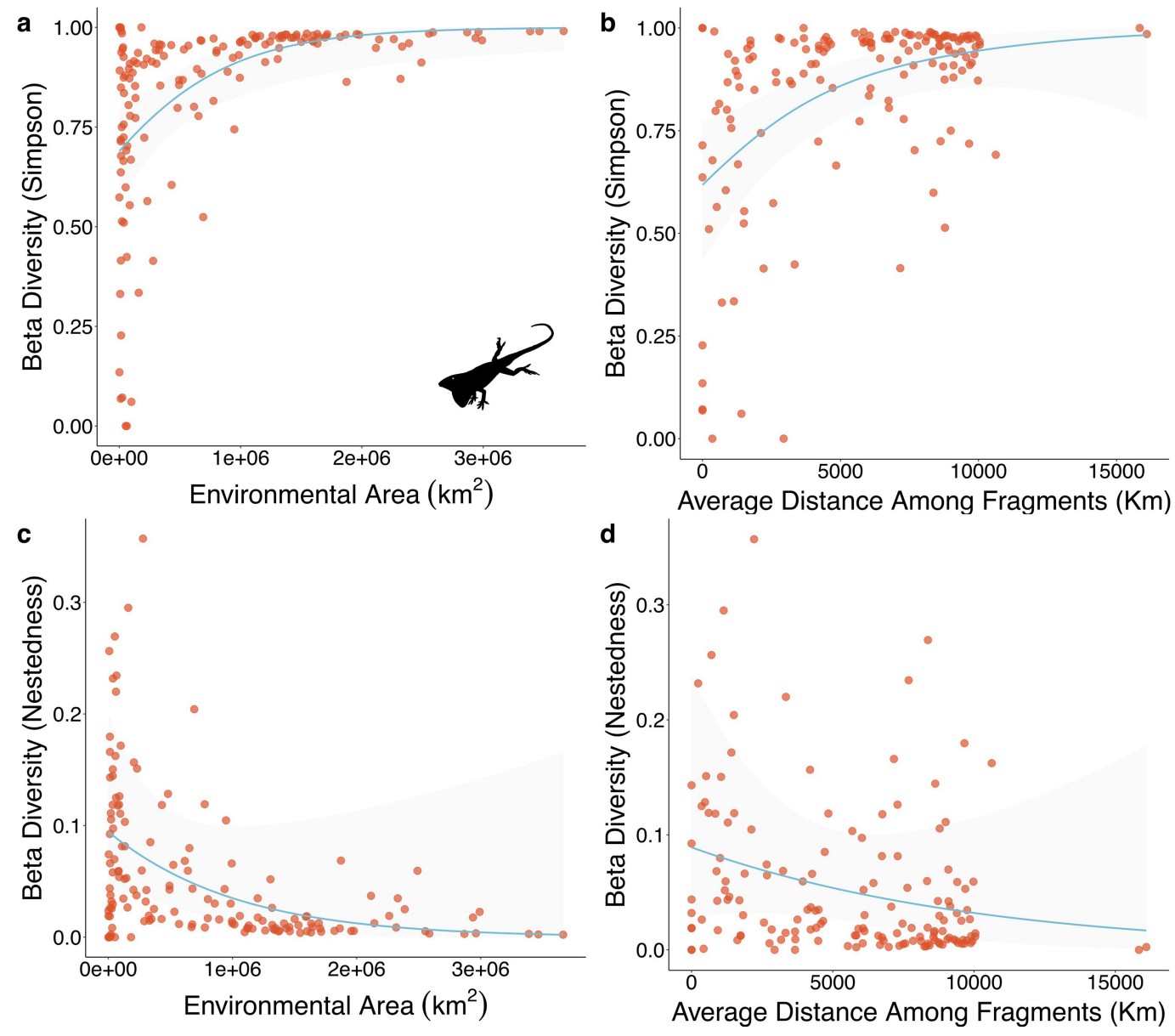

**Extended Data Fig. 8 | The geography of climate and its relationship with community composition for reptiles. a**, **b** the relationship between the geography of climate and the turnover component of beta-diversity. **c**, **d** the relationship between the geography of climate and the nestedness component of beta-diversity (n = 151 climate cells). The grey shading around each regression line represents the 95% confidence interval.

## Extended Data Table 1 | Assessment of Predictor Impact on Tetrapod Richness Patterns Using General Additive Models

*Birds - Adjusted $R^2$ = 0.92, Predicted $R^2$ = 0.92*

| | Proportion of Null Deviance | VIF | Chi.sq | p-value | Chi.sq (accounting for climate autocorrelation) | p-value (accounting for climate autocorrelation) |
|---|---|---|---|---|---|---|
| Climate Area | 0.08 | 1.38 | 17227 | 2.00E-16 | 2513.82 | 2.00E-16 |
| Climate Isolation | 0.07 | 1.22 | 10059 | 2.00E-06 | 5205.7 | 2.00E-16 |
| PC1 | 0.12 | 1.26 | 31528 | 2.00E-16 | 29.24 | 1.47E-16 |
| PC2 | 0.02 | 1.41 | 6226 | 2.00E-16 | 352.16 | 2.00E-16 |
| Joint Contribution of the geography of climate | 0.13 | | | | | |
| Joint Contribution of climate itself | 0.02 | | | | | |
| Joint Contribution between the geography of climate and climate itself | 0.46 | | | | | |
| Total | 0.90 | | | | | |

*Mammals - Adjusted $R^2$ = 0.9, Predicted $R^2$ = 0.92*

| | Proportion of Null Deviance | VIF | Chi.sq | p-value | Chi.sq (accounting for climate autocorrelation) | p-value (accounting for climate autocorrelation) |
|---|---|---|---|---|---|---|
| Climate Area | 0.09 | 1.38 | 7140 | 2.00E-16 | 1108.79 | 2.00E-16 |
| Climate Isolation | 0.05 | 1.22 | 3000 | 2.00E-16 | 1630.08 | 2.00E-16 |
| PC1 | 0.11 | 1.26 | 9864 | 2.00E-16 | 268.31 | 2.00E-16 |
| PC2 | 0.03 | 1.41 | 2560 | 2.00E-16 | 341.89 | 2.00E-16 |
| Joint Contribution within the geography of climate | 0.13 | | | | | |
| Joint Contribution within climate itself | 0.03 | | | | | |
| Joint Contribution between the geography of climate and climate itself | 0.46 | | | | | |
| Total | 0.90 | | | | | |

*Amphibians - Adjusted $R^2$ = 0.90, Predicted $R^2$ =0.91*

| | Proportion of Null Deviance | VIF | Chi.sq | p-value | Chi.sq (accounting for climate autocorrelation) | p-value (accounting for climate autocorrelation) |
|---|---|---|---|---|---|---|
| Climate Area | 0.09 | 1.38 | 5669 | 2.00E-16 | 312.68 | 2.00E-16 |
| Climate Isolation | 0.05 | 1.20 | 2405 | 2.00E-16 | 650.99 | 2.00E-16 |
| PC1 | 0.14 | 1.26 | 9619 | 2.00E-16 | 36.82 | 2.00E-16 |
| PC2 | 0.02 | 1.42 | 1250 | 2.00E-16 | 525.12 | 2.00E-16 |
| Joint Contribution within the geography of climate | 0.12 | | | | | |
| Joint Contribution within climate itself | 0.01 | | | | | |
| Joint Contribution between the geography of climate and climate itself | 0.45 | | | | | |
| Total | 0.88 | | | | | |

*Reptiles - Adjusted $R^2$ = 0.9, Predicted $R^2$ = 0.91*

| | Proportion of Null Deviance | VIF | Chi.sq | p-value | Chi.sq (accounting for climate autocorrelation) | p-value (accounting for climate autocorrelation) |
|---|---|---|---|---|---|---|
| Climate Area | 0.09 | 1.38 | 9237 | 2.00E-16 | 174.5 | 2.00E-16 |
| Climate Isolation | 0.05 | 1.16 | 4246 | 2.00E-16 | 354.4 | 2.00E-16 |
| PC1 | 0.14 | 1.25 | 17053 | 2.00E-16 | 134.9 | 2.00E-16 |
| PC2 | 0.02 | 1.47 | 2116 | 2.00E-16 | 261.8 | 2.00E-16 |
| Joint Contribution within the geography of climate | 0.12 | | | | | |
| Joint Contribution within climate itself | 0.01 | | | | | |
| Joint Contribution between the geography of climate and climate itself | 0.45 | | | | | |
| Total | 0.88 | | | | | |

These models estimate the influence of climate area, climate isolation (geography of climate component), and principal components one and two (climate itself component), on the observed patterns of tetrapod richness. The analysis was conducted on a gridded climate space, defined with 20 equal interval divisions along the climate axis. Each predictor's importance is gauged by the reduction in deviance when excluded from a model that initially included all predictors. The collective influence of predictors is evaluated in three dimensions: (i) the joint effect of geography of climate elements (climate area and climate isolation), (ii) the joint effect of climate itself elements (PC1 and PC2), and (iii) the integrated effect of both geographical and climate elements. Thus, the analysis captures the shared influence within each predictor group (i.e., geography of climate or climate itself) and between the two groups. The p-values for each variable are computed using two-sided tests.

# Reporting Summary

## Statistics

For all statistical analyses, confirm that the following items are present in the figure legend, table legend, main text, or Methods section.

| n/a | Confirmed | |
|---|---|---|
| ☐ | ☒ | The exact sample size (*n*) for each experimental group/condition, given as a discrete number and unit of measurement |
| ☒ | ☐ | A statement on whether measurements were taken from distinct samples or whether the same sample was measured repeatedly |
| ☐ | ☒ | The statistical test(s) used AND whether they are one- or two-sided<br>*Only common tests should be described solely by name; describe more complex techniques in the Methods section.* |
| ☐ | ☒ | A description of all covariates tested |
| ☐ | ☒ | A description of any assumptions or corrections, such as tests of normality and adjustment for multiple comparisons |
| ☐ | ☒ | A full description of the statistical parameters including central tendency (e.g. means) or other basic estimates (e.g. regression coefficient) AND variation (e.g. standard deviation) or associated estimates of uncertainty (e.g. confidence intervals) |
| ☐ | ☒ | For null hypothesis testing, the test statistic (e.g. *F*, *t*, *r*) with confidence intervals, effect sizes, degrees of freedom and *P* value noted<br>*Give P values as exact values whenever suitable.* |
| ☒ | ☐ | For Bayesian analysis, information on the choice of priors and Markov chain Monte Carlo settings |
| ☒ | ☐ | For hierarchical and complex designs, identification of the appropriate level for tests and full reporting of outcomes |
| ☐ | ☒ | Estimates of effect sizes (e.g. Cohen's *d*, Pearson's *r*), indicating how they were calculated |

*Our web collection on statistics for biologists contains articles on many of the points above.*

## Software and code

Policy information about availability of computer code

| Data collection | In the course of this study, we did not utilize any specialized software to collect our data. |
|---|---|
| Data analysis | Data analysis was performed in R version 4.1.2 (2021-11-01) on platform aarch64-apple-darwin20. The code used for statistical analysis is available as supporting material. |

For manuscripts utilizing custom algorithms or software that are central to the research but not yet described in published literature, software must be made available to editors and reviewers. We strongly encourage code deposition in a community repository (e.g. GitHub). See the Nature Portfolio guidelines for submitting code & software for further information.

## Data

Policy information about availability of data

All manuscripts must include a data availability statement. This statement should provide the following information, where applicable:
- Accession codes, unique identifiers, or web links for publicly available datasets
- A description of any restrictions on data availability
- For clinical datasets or third party data, please ensure that the statement adheres to our policy

Global climate and biological data used in our study are open source. All climate data is available at CHELSA (https://chelsa-climate.org/bioclim/) and CGIAR (https://cgiarcsi.community). Vector range maps of all amphibians and mammals are available at IUCN (iucn.org). Bird and squamate range maps are available respectively on Birdlife (Version 2020.1, http://datazone.birdlife.org/species/) and Roll, U. et al. (https://doi.org/10.1038/s41559-017-0332-2).

# Research involving human participants, their data, or biological material

Policy information about studies with [human participants or human data](). See also policy information about [sex, gender (identity/presentation), and sexual orientation]() and [race, ethnicity and racism]().

| | |
|---|---|
| Reporting on sex and gender | Not applicable. |
| Reporting on race, ethnicity, or other socially relevant groupings | Not applicable. |
| Population characteristics | Not applicable. |
| Recruitment | Not applicable. |
| Ethics oversight | Not applicable. |

Note that full information on the approval of the study protocol must also be provided in the manuscript.

# Field-specific reporting

Please select the one below that is the best fit for your research. If you are not sure, read the appropriate sections before making your selection.

☐ Life sciences   ☐ Behavioural & social sciences   ☒ Ecological, evolutionary & environmental sciences

For a reference copy of the document with all sections, see [nature.com/documents/nr-reporting-summary-flat.pdf]()

# Ecological, evolutionary & environmental sciences study design

All studies must disclose on these points even when the disclosure is negative.

| | |
|---|---|
| Study description | In this study, we conducted a global assessment to analyze the influence of climate and geographical characteristics - including fragmentation and size - on the richness and composition of tetrapods. We developed a two-dimensional orthogonal climate space, defined by twelve global scale climate variables, using multiple resolutions from fine to coarse, and considering different definitions of climate space. This enabled us to measure species diversity and compositional dissimilarities for each uniquely classified climate. Each climate was defined by its size, degree of fragmentation and dispersion across the globe, and intrinsic climatic characteristics. These parameters served as predictor variables, which we analyzed in relation to species richness and composition. Our analytical approach involved both linear and non-linear multivariate models, with the non-linear models being tested using different implementation methods. This comprehensive approach allowed us to account for a variety of climatic variables and better understand their effects on tetrapod richness and composition. |
| Research sample | In our study, we first established the presence and absence of various tetrapod groups, specifically amphibians, reptiles, birds, and mammals, across more than 30,000 species, within each classified climate. The climate classification was global, implying that our study also spans a global scale. To execute this classification, we utilized twelve distinct climatic variables that represent the current climate. This approach allowed us to accurately determine the distribution and prevalence of the studied species across different climate classifications worldwide. |
| Sampling strategy | Given the global scale of our research, our approach was designed to encompass a comprehensive scope, ensuring the inclusion of all pertinent data in order to effectively capture the richness and composition of tetrapods across various climatic classifications worldwide. Thus, we utilized all relevant data available to us without predetermining sample size through statistical methods. |
| Data collection | All data for this study were derived from existing datasets, and were collected online by the authors themselves. |
| Timing and spatial scale | The spatial scale of this study is global, with both biodiversity and climate data representing current conditions. As such, our findings provide a snapshot of the present state of tetrapod richness and composition in relation to various climatic classifications worldwide. |
| Data exclusions | No data were excluded from the analyses. |
| Reproducibility | This research does not fall into the category of experimental studies; rather, it is an observational study. Consequently, traditional experimental reproducibility is not applicable. |
| Randomization | This research does not fall into the category of experimental studies; rather, it is an observational study. Consequently, traditional experimental randomization is not applicable. |
| Blinding | This research does not fall into the category of experimental studies; rather, it is an observational study. Consequently, blinding is not applicable. |

Did the study involve field work?   ☐ Yes   ☒ No

# Reporting for specific materials, systems and methods

We require information from authors about some types of materials, experimental systems and methods used in many studies. Here, indicate whether each material, system or method listed is relevant to your study. If you are not sure if a list item applies to your research, read the appropriate section before selecting a response.

## Materials & experimental systems

| n/a | Involved in the study |
|-----|----------------------|
| ☒ ☐ | Antibodies |
| ☒ ☐ | Eukaryotic cell lines |
| ☒ ☐ | Palaeontology and archaeology |
| ☒ ☐ | Animals and other organisms |
| ☒ ☐ | Clinical data |
| ☒ ☐ | Dual use research of concern |
| ☒ ☐ | Plants |

## Methods

| n/a | Involved in the study |
|-----|----------------------|
| ☒ ☐ | ChIP-seq |
| ☒ ☐ | Flow cytometry |
| ☒ ☐ | MRI-based neuroimaging |

