## [Peer Review File · Nature]

Manuscript Title: The geography of climate and the global patterns of species diversity

Reviewer Comments & Author Rebuttals

Reviewer Reports on the Initial Version:

Referees' comments:

Referee #1 (Remarks to the Author):

The authors used a “Grinnellian niche” approach to assess the beta diversity of tetrapod groups according to the geography of climate across the world. To do so, they used the two first PCA axes to synthesize the effects of climate while accounting for land area per climatic unit. Then, using GAM they relate species presence data to explain their geographical patterns. They concluded by proposing to revise the area hypothesis to a more inclusive hypothesis that accounts for both area and climate which they called the area-isolation hypothesis. Overall, the analyses are fine but see some specific comments below, yet I was surprised that they did not comment on the concepts of climate velocity (e.g., Carroll et al. 2017, Brito-Morales et al 2018) in their discussion/conclusion. Indeed, while the proposed research was not about conservation per se, they could have commented at the minimum on the implications of their findings to species conservation under climate change.

Specific comments

Abstract: Lines 21-25: “Species turnover among climates increases with climate area and isolation which can potentially increase speciation rates through capacity rules and reduced gene flow. Across the world, climates occupying larger areas are more fragmented and isolation among similar climatic conditions decreases toward the poles.”

As a reader that reads the abstract first, the terminology of “climate area” and “climate isolation” does not make any sense. Hence, please rewrite the abstract so that people can understand what these terms are as it is the trust of this paper.

Lines 90-96: “We used the first two axes of a principal component analysis (PCA) of twelve global scale climate variables to define a two dimensional orthogonal climate space. These two axes captured ~80% of the global variation in climate and were gridded using equal intervals (Fig1a, Fig1b, Fig1c) to represent thermal (Fig 1c) and water (Fig 1a) availability limits to species distribution. Each cell of the climate space represents multiple geographical cells that fall within a specific climatic interval (Fig 1b) and each geographical cell belongs to a unique climate cell.”

Everything is in the details which are lacking here. Please explain what these grid cells (“gridded using equal intervals”) are, how you decide on their “size”, and why you did not keep the “geographical cells”. All the subsequent analyses/results/interpretations depend on the decisions.

Lines 88: Results and Discussion. The first part of this section read more like a method section. Please move the more technical aspects of the methods to the methods section.

Lines 98-101: "Using this approach we can compute climate area, the sum of the land surface of all geographic cells that occur within a climate cell (Fig. 1f), and climate isolation, represented by the average distance among fragments of the same climate."

Again, it is not clear what is this "climate area" variable and how relevant it is given the arbitrary size of these climate cells. Then, I have much more problems with the second variable "climate isolation": what is a "fragment"? Land within a geographical area? If the geographical areas are ~110km then the effects of mountainous areas on habitat heterogeneity and climatic heterogeneity are smoothed out in many regions of the world.

Figure 1. Please what is "each equal area geographical cell"? Then what is "duality" in the Figure and is it really a duality? Figure 1b: What is the grid size? It seems very arbitrary. Why not use the geographical cells of ~110km?

Figure 4. Please change the colors so that it is clear what is what. Then, please explain what this "Joint Contribution" is? I do not understand what was done here. Then, is it that the Climate Isolation and Climate Area are derived from PC1 and PC2, right? Does this create confounding effects?

Line 340: "Biological Data in Geographical Space." and Line 348: "From Geographical to Climate Space." Please indicate the time periods for these datasets: do they indeed overlap? If not, then all the analyses performed are not relevant.

References

Brito-Morales I, Molinos JG, Schoeman DS, Burrows MT, Poloczanska ES, Brown CJ, Ferrier S, Harwood TD, Klein CJ, McDonald-Madden E, Moore PJ. Climate velocity can inform conservation in a warming world. *Trends in ecology & evolution*. 2018 Jun 1;33(6):441-57

Carroll C, Roberts DR, Michalak JL, Lawler JJ, Nielsen SE, Stralberg D, Hamann A, Mcrae BH, Wang T. Scale-dependent complementarity of climatic velocity and environmental diversity for identifying priority areas for conservation under climate change. *Global Change Biology*. 2017 Nov;23(11):4508-20.

Referee #2 (Remarks to the Author):

This manuscript argues that we should take a geographic approach to looking at species diversity patterns in climate space, in addition to in geographic space. It supports the claim with attractive figures that map e.g. the climate similarity, species diversity in climate space,

If you'll excuse me for making a somewhat frivolous comment, I'm reminded a little of the art piece "Banana taped to wall" by Maurizio Cattelan. The paper takes something that I think most researchers think of as everyday knowledge and puts it on display, which potentially makes the reader look at it in a new way, or learn something new about it. As the authors remark themselves

(line 70): "Using multidimensional space defined by climatic conditions to study biodiversity patterns is not novel...". As an example, the plots in fig. 2 seems to me to be nothing more than a bivariate version of a simple scatter plot of diversity vs a climate variable, as seen in e.g. Currie 1991.

What is new, I believe, is to use metrics from island biogeography (mostly area of land that falls within a climatic grid cell - they also look at isolation, but climatic isolation seems much more well-described), display them, and include them in an analysis. The figure of residuals (Fig 6) displays some interesting patterns, where it seems quite apparent that the pattern in residuals trace out the borders of vegetation/climate biomes. Which could relevantly be considered in the light of evolutionary history or in considering the effect of "transit" regions. Also, the observation that tropical mountains don't represent an outlier here (because I presume the great amount of climate area in a grid cell) is also interesting. And the Fig. 5 on beta diversity components is interesting, though very little space is given to interpret this.

Is this sufficient to make the paper relevant to publish in Nature? I'm a little on the fence - figures are attractive, and I think that insisting on understanding diversity as a pattern in climate space driven by a process in climate space is a take that could lead to interesting theory development.

As it stands, the theory is somewhat undeveloped. What exactly are the predicted mechanisms for climatic isolation and climatic area to drive species diversity? The results as displayed in Fig 4 also seem inconclusive, and I'm a little surprised at the larger joint contribution of orthogonal PCA axes. I also identify a conceptual difficulty (hard to address) in the figures and data assume that movement along any dimension in climate should have the same effect (though Fig 1. d seems almost exclusively driven by PC1). This is an implicit assumption of PCAs in general but the question is how well this is upheld for climate. In particular I can't find information on whether the precipitation-related metrics have been log-transformed prior to analysis - though I believe some kind of normalization might help address this.

Referee #3 (Remarks to the Author):

In this outstanding manuscript, the authors investigate the effects of the geography of climate (climate area and isolation) and the climate itself (temperature, precipitation, etc.) on species richness within tetrapod vertebrates (30,000+ species). Employing state-of-the-art data and statistics, they find that the joint effects of climate, climate area and climate isolation together explain approximately 90% of the variation in tetrapod richness. These results advance our understanding of why some regions and climates harbor dramatically more species than others, one of the classic questions in biology.

The novelty of the study consists in addressing this longstanding question within the new context of the multidimensional climate space and then projecting the results to understand species diversity across geographic regions. Taking this conceptually novel approach, the authors arrive at original conclusions that extend and revise one of the classic hypotheses in the field (i.e. more species occur in larger regions) by showing that the effects geographic area are, when considering the global scale,

inherently intertwined with the effects of geographic isolation (tropics are larger but also more fragmented). In addition to the conceptual novelty, the study develops and presents a new methodology to study diversity patterns in the climate space, which can be easily adapted and implemented by other scientists for their own research and thus will be of potentially broad interest to the research community. Employing their new method, the authors gain significant novel insights into why species diversity varies so dramatically across climates and regions.

The authors use the most up-to-date data on the geographic distributions of species across multiple taxa (mammals, birds, amphibians and reptiles) and state-of-the-art climatic data (temperature, precipitation from CHELSA and potential evapotranspiration from CGIAR) at multiple spatial resolutions. The results will be of immediate interest to the broader research community, spanning multiple disciplines (biogeography, macroecology, community ecology, evolutionary biology, conservation, climate change research). I did not find any flaws that would prohibit publication. All statistical tests are very well executed (correcting for spatial autocorrelation, collinearity, etc.), appropriate, transparently and accurately described, including properly reported error bars, confidence intervals and probability values.

The study is well-written, analyses competently implemented, and the conclusions highly original and of broad interest. I particularly appreciated the methodological approach toward working within the multidimensional climate space. The climate space and the geographic isolation are notoriously hard to capture, but the presented methodology addresses these challenges in a compelling, yet elegantly simple way. The discussion of the results is balanced and brilliantly connects the findings to classic literature (Terborgh, MacArthur and Wilson), but also to recent studies, while considering multiple perspectives across fields (historical contingency as the driver of richness patterns, effects of past climates, mountains as refuges), with potentially important new lessons for biodiversity conservation (changes in climate area and isolation in the near future).

I would not suggest any major revisions. Perhaps I have only one minor recommendation. Namely, can the authors elaborate on their results when they change the definition of the climate space? For example, would the conclusions change if the climate space included only temperature and precipitation (which define biomes in the Whittaker diagram) or when removing evapotranspiration or adding environmental productivity? The main conclusions are very well-supported, and I do not expect them to change, given their robustness across taxa and toward different measures of climate area and isolation. But it could be useful for the reader and future studies, which will likely adopt this methodology, to elaborate briefly on the construction of the climate space within the main text of the manuscript (e.g. adding a paragraph or a couple of sentences).

Author Rebuttals to Initial Comments:

Revision overview

We would like to thank you for considering our manuscript for publication in Nature and for giving us the opportunity to revise it. The insightful feedback from the three referees has significantly enhanced the clarity and rigor of our study. The comments raised by the referees were related to: (i) clarifying the definitions of some concepts; (ii) expanding our discussions about the mechanisms underlying our findings and about the application of our approach to conservation biology; and (iii) sensitivity analysis to different combinations of variables used to define climate space.

Overall, *Referee #1* suggested: adding a discussion about the implications of our study to conservation biology (lines 83-84, lines 306-329), editing in the abstract (lines 15-30), increasing details in methodological procedures (lines 89-109) and changing colors in the figures (lines 176-183), all of which were performed. *Referee #2* suggested: adding more details about the mechanisms underlying our findings (lines 186-199), increasing details in the interpretation of results (lines 241-252), clarifying the explanation of some results (lines 155-158) and conducting sensitivity analyses regarding transformations of some climate variables (lines 468-473 and Supplementary information), all of which were performed. Finally *Referee #3* suggested conducting a sensitivity analysis associated with different definitions of climate space (i.e. climate space defined with different variables) which is presented (lines 468-473) in the revised manuscript (Supplementary information). All sensitivity analyses show that our results are very consistent, independent of the definition of climate space. Below you can find our detailed point by point responses to Referee's comments:

Point by point response to Referee #1:

Referee #1:

"The authors used a "Grinnellian niche" approach to assess the beta diversity of tetrapod groups according to the geography of climate across the world. To do so, they used the two first PCA axes to synthesize the effects of climate while accounting for land area per climatic unit. Then, using GAM they relate species presence data to explain their geographical patterns. They concluded by proposing to revise the area hypothesis to a more inclusive hypothesis that accounts

for both area and climate which they called the area-isolation hypothesis. Overall, the analyses are fine but see some specific comments below, yet I was surprised that they did not comment on the concepts of climate velocity (e.g., Carroll et al. 2017, Brito-Morales et al 2018) in their discussion/conclusion. Indeed, while the proposed research was not about conservation per se, they could have commented at the minimum on the implications of their findings to species conservation under climate change”.

We thank the reviewer for investing his/her time in carefully reading and commenting on our manuscript. We agree that we did not expand on the potential implications of our findings to conservation under climate change, which was only briefly mentioned in the previous version of the manuscript. In the current version of the manuscript we now discuss these implications following the suggestions of the referee (Lines 83-85). We also comment on climate change velocity arguing that velocity measures should not only account for how fast will climate changes in certain regions but also for the speed of changes in commonness and isolation of climatic conditions (Lines 314, 329).

In the context of the effects of the geography of climate on the maintenance and emergence of biodiversity, climate change can lead to expansion and contraction of the geographical extent of climatic conditions and to increase and decrease in connections or isolation among similar climatic conditions. If many species have their optimum climate within a climatic condition that contracts its geographical extent with climate change, then competition might increase, potentially influencing species coexistence. In addition, the contraction of the geographical extent of a climatic condition imposes evolutionary pressure for individuals to shift their optimum to other similar climatic conditions that: (i) are either less saturated with species or, (ii) expanded their geographical extent on the surface of the planet. Finally, connections and isolations of climate affect the degree of isolation among populations and consequently the gene flow among populations (Lines 224-233). Therefore, the effects of climate change on the geography of climate can affect processes related to maintenance and emergence of biodiversity. Even though ecologists and evolutionary biologists are not yet used to thinking about the geographical extent and isolation of climatic conditions, these key properties of the geography of climate should be taken into consideration when trying to understand and mitigate the effects of climate change on biological phenomena.

Referee #1:

“Specific comments

Abstract: Lines 21-25: “Species turnover among climates increases with climate area and isolation which can potentially increase speciation rates through capacity rules and reduced gene flow. Across the world, climates occupying larger areas are more fragmented and isolation among similar climatic conditions decreases toward the poles.”

As a reader that reads the abstract first, the terminology of “climate area” and “climate isolation” does not make any sense. Hence, please

rewrite the abstract so that people can understand what these terms are as it is the trust of this paper”.

We agree with the reviewer. We entirely re-wrote the abstract (Lines 15-30) following the guidelines of Nature in terms of organizing the information in the abstract.

Referee #1:

“Lines 90-96: “We used the first two axes of a principal component analysis (PCA) of twelve global scale climate variables to define a two dimensional orthogonal climate space. These two axes captured ~80% of the global variation in climate and were gridded using equal intervals (Fig1a, Fig1b, Fig1c) to represent thermal (Fig 1c) and water (Fig 1a) availability limits to species distribution. Each cell of the climate space represents multiple geographical cells that fall within a specific climatic interval (Fig 1b) and each geographical cell belongs to a unique climate cell.”

Everything is in the details which are lacking here. Please explain what these grid cells (“gridded using equal intervals”) are, how you decide on their “size”, and why you did not keep the “geographical cells”. All the subsequent analyses/results/interpretations depend on the decisions”.

We agree with the referee. We performed several editions in the main text and in Fig 1’s legend to clarify what the reviewer pointed out (Lines 89-109). The size of the grid cell in each one of these spaces are differently defined because in geographical space the two axes used to define a gridded space are longitude and latitude while in climate space the two axes are climatic variables.

Gridded geographical space - In the two dimensional (x = longitude and y = latitude) geographical space, a gridded map is used to summarize the presence and absence of species for tetrapods groups. The size of each grid cell is defined as ~110 km (110km intervals in longitude and latitude axis) resolution with Behrmann equal area projection (approximately 1° resolution) (Lines 443-450). The choice of this resolution is based on empirical studies that explored the ideal spatial resolution for different biological data. Therefore, the resolution cited above is the most appropriate for the type of data used in our study as detailed in the methods section (Lines 560-563).

Gridded climate space - In the two dimensional climate space (x = PC1 and y = PC2) a gridded map is used to compute the geographical extent and isolation of climatic conditions as well as to count the number of species (i.e. species richness) occurring in each climatic condition (climate grid cell). The size of each climate grid cell is defined based on intervals in each climate variable (lines 447-490). In Fig 1b of the main text (inserted below), the cloud of blue points define the observed climate. A grid is put above the observed climate and the size of cells is defined by intervals in each axis. To have climate cells of the same size in climate space *equal intervals* divide each axis and define the limits of grid cells creating therefore a grid:

To deal with the uncertainty of the cell size choice we analyzed five different resolutions of climate space which is referred by referee #3 as one the strengths of our study.

In the figure below for example, which is part of Fig S2 of the supplementary information, we mapped bird richness in four resolutions beyond the one shown in the main text:

The cell size decreases from the first figure on the left to the last figure on the right. By varying the size of climate cells from coarse to finer definitions of climate we demonstrate that mapping and studying biodiversity patterns in all these different resolutions show consistency in results not only across resolutions of climate space but also across tetrapods groups (Lines 485-486, Table S2 to Table S5, Supplementary information).

Referee #1:

“Lines 88: Results and Discussion. The first part of this section read more like a method section. Please move the more technical aspects of the methods to the methods section”.

We thank the reviewer for pointing this out. We edited this paragraph to reduce the methodological detail but we decided not to completely remove it for several reasons. Some methodological details should appear in the results and discussion session to ensure that readers are not confused about crucial methodological procedures. In addition, given that the concept/approach is new, many readers will be unfamiliar with it and will likely have difficulty interpreting the results. Finally, we were also asked to include additional methodological detail by referee #1 (see previous comment) so we tried to balance these additions with reducing some details but keeping the essential. Therefore, we prefer to leave minimal methodological detail in the text to increase the accessibility and readability of the results and discussion. Minimum methodological details are presented in lines 89 to lines 109.

Referee #1:

“Lines 98-101: “Using this approach we can compute climate area, the sum of the land surface of all geographic cells that occur within a climate cell (Fig. 1f), and climate isolation, represented by the average distance among fragments of the same climate.”

Again, it is not clear what is this “climate area” variable and how relevant it is given the arbitrary size of these climate cells. Then, I have much more problems with the second variable “climate isolation”: what is a “fragment”? Land within a geographical area? If the geographical areas are ~110km then the effects of mountainous areas on habitat heterogeneity and climatic heterogeneity are smoothed out in many regions of the world”.

We thank the reviewer for pointing this out. We clarify the explanation of how the area of a climatic condition is computed (Lines 98-99). And we also clarify what is referred to as a climate fragment and how it can be calculated (Lines 100-101). The comments about the resolution goes in the same direction of answers we gave above and that are already solved in the main text. It is important to note here that there are two types of resolutions that are defined in our study as mentioned in the previous responses. One is the spatial resolution which defines how biological data is mapped in geographical space. This resolution is set ~110km not arbitrarily but based on empirical studies that explore and define the best resolution for the type of data that is used in this study (Lines 443-449). As the reviewer noticed however, indeed this type of resolution might not be enough to capture the effect of mountainous climate across the world and we explicitly mentioned this in our discussion (Lines 282-283). Even though finer resolutions would be ideal for mountainous regions, mountainous regions of the world appear in the residual patterns of tetrapods groups (Lines 281-292, Fig 6) which can still lead to interesting discussions of smaller size of mountainous climate not being capable of explaining their immense biodiversity (Lines 281-292). The matter of which spatial resolution to use to better capture geographical characteristics of climate depends exclusively on what is the best spatial resolution for the taxonomic groups in focus and the data used to represent their distribution. Therefore we would expect that scientists applying the same framework of our study to other groups of organisms will not only define different resolutions than 110km if distributional range data is not the focus, but will also define climate space with different axis as the same climatic data used in our study of terrestrial tetrapods would not be ideal to study aquatic organisms, for example.

The other type of resolution in our study is the resolution in climate space, in which we tested five different resolutions as explained in other responses above and now better detailed in the current version of the manuscript.

Referee #1:

“Figure 1. Please what is “each equal area geographical cell”? Then what is “duality” in the Figure and is it really a duality? Figure 1b: What is the grid size? It seems very arbitrary. Why not use the geographical cells of ~110km”?

We clarify the terms mentioned by the reviewer in figure 1b (Lines XX). The questions raised by the reviewer were partially clarified in other responses above. Equal area geographical cells refers to each geographical cell (i.e. grid cells from geographical space). These cells are equal in their area because of the projection that was used (i.e. Behrmann equal area projection). (Lines 446-447). Mapping simply longitude and latitude cells in a rectangular projection would create cells of decreasing area from the equator to poles and would create difficulties in comparing biodiversity patterns across the globe. Equal-area projections (such as Behrmann’s) are a mean to conserve the area per grid cell.

As answered in a previous comment made by the reviewer there are two different grid size definitions. The choice of grid size in geographical space has empirical support (Lines 443-449) and for climate space we defined five different sizes of grid cells. The conclusions are the same independent of the choices in the size of climate space cells (Lines 474-490, Supplementary information).

The duality concept was also clarified in the manuscript (Lines xx). The duality between geographical and climate space refers to the relationship between geographical and climate space: a given climatic condition (climate cell in climate space) is observed in several geographical locations ($1 \rightarrow n$: one climate cell represent n geographical cells) while several geographical locations belong to a unique climate condition ($n \rightarrow 1$: n geographical cells represent one climate cell). When the term duality is used in the text it is now immediately explained to avoid any potential confusion (Lines 93-96, Lines 120-123)

Referee #1:

“Figure 4. Please change the colors so that it is clear what is what. Then, please explain what this “Joint Contribution” is? I do not understand what was done here. Then, is it that the Climate Isolation and Climate Area are derived from PC1 and PC2, right? Does this create confounding effects?”

Following the suggestion of the reviewer we changed the colors of Fig 4. The previous figure showed only different colors in green tones only, which we particularly were more fond of. In the current version, geography of climate and climate itself have different tones:

The joint contribution mentioned by the referee refers to the amount of variation in species richness that is simultaneously explained by the four predictor variables (climate area, climate isolation, PC1 and PC2) and cannot be attributed to each variable individually. This is now better explained in the current version of the manuscript (Lines 155-158).

Regarding the potential confounding effect mentioned by the reviewer, we found no confounding effect among these variables. The variance inflation factor is very low in the model that includes all of these variables (variance inflation factor < 1.5) which indicates a low correlation among these variables. Indeed, if each of these variables are confronted, the largest correlation among them is below 0.5, which is much lower than the statistical consensus of potential confounding effects among variables $|r| > 0.7$ (Dormann et al., 2013). See correlation matrix below:

Thus, the correlation between the geography of climate and climate itself is very low, which makes it possible to robustly add these variables in the same statistical model. It is important to note that the area of a climatic condition or how isolated it is on the surface of the planet does not give any information about the climate per se (i.e., how warm and wet is a given location). As we can see in Fig 1e and Fig 1f (Lines 123-125), common climates are found in different regions of climate space, as well as isolated climates. Thus, a confounding effect would appear in statistical terms, only if these variables were strongly associated with each other, which is not the case here (518-521). We refer to the absence of relationship among these variables in lines 518 to 521 and show their weak association as supplementary information (Fig S13).

Dormann, C. F. et al. Collinearity: a review of methods to deal with it and a simulation study evaluating their performance. *Ecography* 36, 27–46 (2013).

Referee #1:

“Line 340: “Biological Data in Geographical Space.” and Line 348: ‘From Geographical to Climate Space.’ Please indicate the time periods for these datasets: do they indeed overlap? If not, then all the analyses performed are not relevant”.

Both climatic and biological data represent current climate and current distribution of species (Line 446, Line 461).

Referee #1:

“References

Brito-Morales I, Molinos JG, Schoeman DS, Burrows MT, Poloczanska ES, Brown CJ, Ferrier S, Harwood TD, Klein CJ,

McDonald-Madden E, Moore PJ. Climate velocity can inform conservation in a warming world. Trends in ecology & evolution. 2018 Jun 1;33(6):441-57

Carroll C, Roberts DR, Michalak JL, Lawler JJ, Nielsen SE, Stralberg D, Hamann A, Mcrae BH, Wang T. Scale-dependent complementarity of climatic velocity and environmental diversity for identifying priority areas for conservation under climate change. Global Change Biology. 2017 Nov;23(11):4508-20".

We thank the reviewer for pointing out these important references in the field. We are citing one of these references when referring to climate change velocity (Lines 317)

Point by point response to Referee #2:

Referee #2

This manuscript argues that we should take a geographic approach to looking at species diversity patterns in climate space, in addition to in geographic space. It supports the claim with attractive figures that map e.g. the climate similarity, species diversity in climate space,

If you'll excuse me for making a somewhat frivolous comment, I'm reminded a little of the art piece "Banana taped to wall" by Maurizio Cattelan. The paper takes something that I think most researchers think of as everyday knowledge and puts it on display, which potentially makes the reader look at it in a new way, or learn something new about it. As the authors remark themselves (line 70): "Using multidimensional space defined by climatic conditions to study biodiversity patterns is not novel...". As an example, the plots in fig. 2 seems to me to be nothing more than a bivariate version of a simple scatter plot of diversity vs a climate variable, as seen in e.g. Currie 1991.

We thank the reviewer for investing his/her time in carefully reading and commenting on our manuscript. The comments were very valuable and helped us to increase the clarity and quality of our study.

Unfortunately, we lack the expertise to interpret or judge the value of the art piece mentioned. However, we believe the main point the referee raised is that by looking at something quite ordinary in a new way we can learn something new. The new lenses

provided by our study (biodiversity analysis in the climatic space) reveal important theoretical novelty hidden behind the oldest pattern described in ecology (large scale spatial patterns in biodiversity), even when we are asking one of the oldest questions in natural sciences (i.e. what causes large-scale variation in biodiversity). The hidden effects of the geography of climate on species diversity has been neglected in the body of research in ecology and evolutionary biology. Our new perspective even calls for an extension and revision of the traditional body of biogeographical theory.

In our study, classical concepts in ecology and evolutionary biology, such as the effect of area and isolation on biological processes, are explored in the context of climate. In biological sciences researchers are used to (i) study and investigate the effect of area and isolation on biological patterns as well as the (ii) direct and indirect effects of climatic conditions on biological patterns. By mapping biodiversity patterns in climate space, and disentangling the effect of the geography of climate (i.e. climate area and isolation) and climate itself, we reveal the hitherto hidden effects of the geography of climate on species diversity, such as the degree of geographic fragmentation of similar climate conditions. We also show that, contrary to common wisdom, tropical climate is substantially more disconnected than temperate climate. In addition, as highlighted by reviewer #1, our findings imply that there are aspects of climatic conditions rarely explored in the context of climate change. However, these potential changes in very basic properties of climate have not been the focus of studies in conservation biology because area and isolation are not commonly explored in the context of climate change. Thus, back to the art piece, we agree that there is, and there will always be, a lot to learn by looking from another angle to what was thought to be ordinary.

However, we must clarify that there **is a fundamental difference** between “simple scatter plots of diversity and climate variables” (as seen in Currie 1991) and the figure 2 in our study referred to by the referee. First, the unit of analysis of classic richness-climate scatterplots (such as fig. 3 of Currie 1991) is a location in geographic space. Thus, each dot in those scatterplots, such as those in Currie 1991, corresponds uniquely to a single locality on the globe, which has a corresponding species richness value (usually the vertical axis) and a climatic condition (usually the horizontal axis). This geographic approach to species richness is intuitive and has served macroecology and biogeography for decades. However, there is **no analogy** between fig. 3 in Currie 1991 and fig. 2 of our study. Fig 2 in our study is more analogous to a map than a scatterplot, and this is why we frequently refer to our framework of analysis as “climate space”.

The units of analysis in figure 2 are climatic conditions. Thus, each square in our figure 2 represents a unique and narrow range of climatic conditions (e.g. warm-wet, or cold-dry), which may exist in none, one or multiple geographic regions, all with the same climate but distributed anywhere around the globe. These geographic regions may be adjacent (geographically connected), or disjointly scattered around the globe, with important consequences for the biogeography of species (e.g. geographic isolation). The **geographic distribution** of similar climatic conditions is what we define as the **geography of climate**, and it is one of the main foci of our study.

We understand that, because our fig. 2 has horizontal and vertical climate axes, and the colors represent species richness, one may initially think that the figure is just a bivariate plot of richness-climate. However, using the standard geographic space framework, each point in such a richness-climate bivariate plot would represent a different location in space (such as in fig. 3 of Currie 1991). In addition, each point would have an assigned color

according to its species richness observed in the location, which is calculated based on the number of overlapping distribution of species ranges on the referential geographic area. So, in such a bivariate plot using the geographic space framework, there could be two points located very close to one-another (i.e. similar climates), but with very different species richness and/or composition (e.g. different faunas or floras). In other words, such a plot would still differentiate between regions of the globe with similar climate, as long as they have different species richness and composition, because the framework of analysis is the geographic space. However, in our fig. 2, which has climatic conditions as units of analysis, all regions of similar climate constitute a single unit of analysis (climate space framework). Thus, each point (square) in our fig. 2 not only represents a unique climatic condition, but it also combines in a single unit of analysis all species that exist under the same climatic condition, regardless if these species are distributed continuously in space or across different continents. **Therefore, biodiversity patterns emerging from analyses in geographic vs climatic space are bound to be statistically and conceptually distinct. The results of our statistical models of biodiversity analysis in the climate space (e.g. fig. 4), and our main findings, are absolutely novel, and could not be replicated using the standard framework of analysis in macroecology (geographic space).**

In summary, while fig. 3 in Currie 1991 has **geographic space** as a frame of reference, fig. 2 in our study has **climate space** as a frame of reference. In fact, **the geography of the climate is NOT the focus of Currie 1991. The shift in perspective from geographic space to climatic space, and the theoretical consequences of that shift, is the core innovation of our study.**

Currie, D. J. Energy and Large-Scale Patterns of Animal- and Plant-Species Richness. *The American Naturalist* 137, 27–49 (1991).

Referee #2

What is new, I believe, is to use metrics from island biogeography (mostly area of land that falls within a climatic grid cell - they also look at isolation, but climatic isolation seems much more well-described), display them, and include them in an analysis. The figure of residuals (Fig 6) displays some interesting patterns, where it seems quite apparent that the pattern in residuals trace out the borders of vegetation/climate biomes. Which could relevantly be considered in the light of evolutionary history or in considering the effect of "transit" regions. Also, the observation that tropical mountains don't represent an outlier here (because I presume the great amount of climate area in a grid cell) is also interesting. And the Fig. 5 on beta diversity components is interesting, though very little space is given to interpret this.

We thank the reviewer for raising some interesting points here. Indeed the beta diversity component was given very little space in the previous version of the manuscript which is now

expanded (Lines 241-252). The residual patterns discussed in our text were also expanded to include the point raised by the reviewer about the border of biomes.

Referee #2

Is this sufficient to make the paper relevant to publish in Nature? I'm a little on the fence - figures are attractive, and I think that insisting on understanding diversity as a pattern in climate space driven by a process in climate space is a take that could lead to interesting theory development.

Beyond the points highlighted in the answers above we would like to highlight that the potential to lead to interesting theory development is indeed something that might be immeasurable in the present moment. Putting it in a general context, ecological and evolutionary questions related to the emergence and maintenance of biodiversity, assembly of communities, diversity of functions and organism distributions usually invoke hypotheses formulated in terms of environmental conditions. However, common practices employed by ecologists and evolutionary biologists to evaluate environmental hypotheses and assess potential causes underlying biodiversity patterns can be limited by the space in which statistical relationships are explored. Environmental effects on biodiversity patterns have been studied almost entirely in geographical space which has served well our understanding of how the amount and variability of energy and water availability and their resulting ecosystem productivity are associated with biodiversity patterns. But if the surrounding environment is key to many biodiversity patterns, why should the space where biodiversity is mapped and studied be restricted to the geographical axis alone? **As we show in the current study, many fields in ecology and evolutionary biology might achieve different perspectives and unravel different hidden effects of the environment if the mapping and exploration of biodiversity patterns are done in climate space in addition to geographical space.**

Referee #2

As it stands, the theory is somewhat undeveloped. What exactly are the predicted mechanisms for climatic isolation and climatic area to drive species diversity? The results as displayed in Fig 4 also seem inconclusive, and I'm a little surprised at the larger joint contribution of orthogonal PCA axes.

We expanded the discussion about the predicted mechanisms for climate area (Lines 186-199) and isolation (Lines 224-233). The mechanism underlying species area relationship in climate is different from the common expectations discussed in geographical systems as similar climatic conditions are not continuously distributed across geography but scattered within and across continents. These scattered climatic conditions that sum up to define a total area represent multiple geographical locations within the same type of environment. While in a geographical system a continuous large area naturally leads to higher environmental

heterogeneity and consequently different species exploring different ecological opportunities, the same rationale is not expected to be a strong case in climate space. Within a given climate, there is little environmental heterogeneity that could lead to different specializations for different environmental conditions. Therefore, more species occurring within homogenous climatic conditions occupying larger geographical extents likely emerge as a result of capacity rules where climatic conditions that occupy more land surface support more individuals, larger populations, and consequently increased speciation and decreased extinction rates. The mechanism underlying the association between species richness and climate isolation is likely linked to climate isolation influencing gene flow among diverging populations. Over deep time, at global scale, climate change, continental drift and mountain uplift can affect the spatial connection among similar climatic conditions. Populations of a species dispersing to follow their optimum climatic conditions are expanding, contracting and fragmenting their geographical distribution within climatic conditions. The isolation of populations within spatially disconnected climatic conditions increases the changes of allopatric speciation events and long distance dispersal causing founder speciation events.

Regarding the joint contribution of the orthogonal PCA axis, the joint contribution in Fig 4 is not only from the orthogonal PCA axis but from climate (first two principal components) and the geography of climate (climate area and climate isolation). Now better explained in the manuscript (Lines 155-158): Approximately 60% of the variation in species richness is simultaneously explained by climate itself and the geography of climate and cannot be attributed to each variable individually (i.e. variables joint contribution) which suggests non trivial answers to the effects of climate.

I also identify a conceptual difficulty (hard to address) in the figures and data assume that movement along any dimension in climate should have the same effect (though Fig 1. d seems almost exclusively driven by PC1). This is an implicit assumption of PCAs in general but the question is how well this is upheld for climate.

We believe some confusion here is caused by Fig 1d that represents just a color scheme to illustrate the duality between climate and geographical space. This figure does not show any result or empirical pattern and is used for didactic purposes to explain the duality between climate and geographical space. Attributing colors to one climate cell (Fig 1d) will give the same colors in several geographical cells that belong to this climate (Fig 1e). The choice to use this type of color gradient is that contrasting zones in climate space can be easily identified by the readers. As we can see below broad climatic zones ranging from tropical to extratropical regions can be identified in the figure (Fig 1d and Fig 1e) chosen to compose the main text:

d

e

Any type of gradient could be chosen to represent the same figure. For example, instead of a color pattern spatially structured in climate space, random colors could be attributed to climate space and reprojected to geographical space:

Or a color gradient varying only in the vertical axis:

Or a color gradient established in a different diagonal than the one chosen in Fig 1d and Fig1e.

Therefore, in Fig 1d, PC1 is not driving any relevant pattern as mentioned by the reviewer. How the color gradient is displayed is a decision to better illustrate the relationship between climate and geographical space and to help readers identify broader climate zones when looking at climate space. We now explain in the figure legend that Fig 1d is used to illustrate the concept of the duality between climate and geographical space which allows geographical information to be mapped in climate space and vice-versa (Lines 114-125). As supplementary material we included a supplementary figure with four different color gradients (Fig S12, Supplementary information).

In particular I can't find information on whether the precipitation-related metrics have been log-transformed prior to analysis - though I believe some kind of normalization might help address this.

The precipitation variables mentioned by the reviewer have a skewed distribution and become more symmetrical after log transformation. However, their transformation previous to the PCA analysis do not affect the patterns observed for both principal components. The correlation among principal components that assumed the transformed variables and the principal components that used all variables in its original scale is very close to 1 (0.98 and 0.96 for the first two principal components) as we can see in the figure below:

The results are very stable because the PCA is computed in a correlation matrix given that the twelve variables representing limits in energy and water availability are in different scales. When these variables are summarized in completely orthogonal axes, the previous transformation of some of these variables does not affect their variability across space and therefore has no effect in the principal components used in our study. We ran all the analysis log transforming the precipitation variables and the results are very robust to these transformations (Table S15 to Table S18). In addition, results for environmental space defined with other variables are also shown following referee #3 suggestions (Table S11 to Table S14, Supplementary information). Below we can see one example for birds of how the patterns are very similar if the transformations were performed.

Part of table S15 | Observed statistics of general additive models for birds. Models were fitted assuming the richness patterns of birds as response variable and climate area, climate isolation and principal components one and two as predictors. Here climate space is defined by the first two axes of a PCA with the twelve climate variables used in the main text, **but prior to the PCA, precipitation variables were log transformed.**

Birds				
	Percentage of Variance	VIF	Chi.sq	p-value
Climate Area	0.062	1.12	15778	<2E-16
Climate Isolation	0.041	1.73	10001	<2E-16
PC1	0.123	1.41	34638	<2E-16
PC2	0.024	1.19	6228	<2E-16
Joint Contribution	0.669			

Point by point response to Referee #3:***Referee #3 (Remarks to the Author):***

In this outstanding manuscript, the authors investigate the effects of the geography of climate (climate area and isolation) and the climate itself (temperature, precipitation, etc.) on species richness within tetrapod vertebrates (30,000+ species). Employing state-of-the-art data and statistics, they find that the joint effects of climate, climate area and climate isolation together explain approximately 90% of the variation in tetrapod richness. These results advance our understanding of why some regions and climates harbor dramatically more species than others, one of the classic questions in biology.

The novelty of the study consists in addressing this longstanding question within the new context of the multidimensional climate space and then projecting the results to understand species diversity across geographic regions. Taking this conceptually novel approach, the authors arrive at original conclusions that extend and revise one of the classic hypotheses in the field (i.e. more species occur in larger regions) by showing that the effects geographic area are, when considering the global scale, inherently intertwined with the effects of geographic isolation (tropics are larger but also more fragmented). In addition to the conceptual novelty, the study develops and presents a new methodology to study diversity patterns in the climate space, which can be easily adapted and implemented by other scientists for their own research and thus will be of potentially broad interest to the research community. Employing their new method, the authors gain significant novel insights into why species diversity varies so dramatically across climates and regions.

The authors use the most up-to-date data on the geographic distributions of species across multiple taxa (mammals, birds, amphibians and reptiles) and state-of-the-art climatic data (temperature, precipitation from CHELSA and potential evapotranspiration from CGIAR) at multiple spatial resolutions. The results will be of immediate interest to the broader research community, spanning multiple disciplines (biogeography, macroecology, community ecology, evolutionary biology, conservation, climate change research). I did not find any flaws that would prohibit publication. All statistical tests are very well executed (correcting for spatial autocorrelation, collinearity, etc.),

appropriate, transparently and accurately described, including properly reported error bars, confidence intervals and probability values.

The study is well-written, analyses competently implemented, and the conclusions highly original and of broad interest. I particularly appreciated the methodological approach toward working within the multidimensional climate space. The climate space and the geographic isolation are notoriously hard to capture, but the presented methodology addresses these challenges in a compelling, yet elegantly simple way. The discussion of the results is balanced and brilliantly connects the findings to classic literature (Terborgh, MacArthur and Wilson), but also to recent studies, while considering multiple perspectives across fields (historical contingency as the driver of richness patterns, effects of past climates, mountains as refuges), with potentially important new lessons for biodiversity conservation (changes in climate area and isolation in the near future).

I would not suggest any major revisions. Perhaps I have only one minor recommendation. Namely, can the authors elaborate on their results when they change the definition of the climate space? For example, would the conclusions change if the climate space included only temperature and precipitation (which define biomes in the Whittaker diagram) or when removing evapotranspiration or adding environmental productivity? The main conclusions are very well-supported, and I do not expect them to change, given their robustness across taxa and toward different measures of climate area and isolation. But it could be useful for the reader and future studies, which will likely adopt this methodology, to elaborate briefly on the construction of the climate space within the main text of the manuscript (e.g. adding a paragraph or a couple of sentences).

We are very honored to receive so many compliments about the quality of our study. We thank the reviewer for investing time in carefully reading and commenting on our manuscript.

The problem we see when using only two climatic variables, such as in the Whittaker diagram mentioned by the reviewer, is the strange definition of space that it creates, given the existent relationship among temperature and precipitation. In geographical space we use artificial variables such as latitude and longitude to define a completely independent two dimensional space. The independence of these axes allows, for example, us to establish perfectly independent grid cells across the world. However, the correlation between temperature and precipitation makes things conceptually more complicated because it implies that the axes used to define a space are not independent axes. The mathematical solution for this problem is to work with orthogonal axes which can be obtained through a PCA. The PCA also allows the inclusion of several variables that are important to represent variations in energy and water availability, especially not only average conditions but also limiting conditions such as maximum and minimum temperature and precipitation. Thus, having orthogonal axes to represent a new space where biodiversity patterns can be mapped and studied is the ideal option in our opinion. We expect that if the same approach is taken for

different organisms, for example aquatic organisms, the choice of variables will off-course change.

Nonetheless, we added the sensitivity analysis requested which are now mentioned in the main text (Lines 469-474). As requested by the reviewer we also added a sensitivity analysis for a space in which PET is replaced by NPP. As NPP is a result of energy and water availability, the conclusions remain the same.

Bellow we can see one example for birds of how the results are robust if PET is replaced by NPP:

part of table S7 | Observed statistics of general additive models for birds. Models were fitted assuming the richness patterns of birds as response variable and climate area, climate isolation and principal components one and two as predictors. **Here NPP replaces PET in the PCA** as explained in Sensitivity analysis for different definitions of the climate space section in supplementary information.

Birds				
	Percentage of Variance	VIF	Chi.sq	p-value
Climate Area	0.092	1.36	19851	<2E-16
Climate Isolation	0.060	1.24	11937	<2E-16
PC1	0.152	1.29	34057	<2E-16
PC2	0.021	1.31	4631	<2E-16
Joint Contribution	0.564			
Total	0.889			

The robustness is also observed if principal components (Using PET or NPP) are confronted (pearson's r ca. 0.96) as in shown in the figure below:

Fig S9 | Sensitivity Analysis for the choice of variables in the PCA. Here the first two principal components of two different PCAs are confronted. PC1 and PC2 in y axis represent the scores of a PCA in which PET was replaced by NPP. On the other hand, PC1 and PC2 in the x axis represent the scores of a PCA in which NPP was replaced by PET. **A** PC1 emerging from the two different PCAs are confronted (pearson's $r = 0.953$) and **B** PC2 emerging from the two different PCAs are confronted (pearson's $r = 0.954$).

Finally we can also see that the results are robust if climate space is defined by temperature and precipitation. However, we emphasize again that the relationship between temperature and precipitation implies that the axes used to define a “space” are not independent.

Part of table S11 | Observed statistics of general additive models for birds. Models were fitted assuming the richness patterns of birds as response variable and climate area, climate isolation and mean annual temperature and mean annual precipitation as predictors. Here climate space is defined by temperature (Bio1) and precipitation (Bio12) as explained in Sensitivity analysis for different definitions of the climate space section (Supplementary information).

Birds				
	Percentage of Variance	VIF	Chi.sq	p-value
Climate Area	0.083	1.81	23015	<2E-16

Climate Isolation	0.051	1.25	13158	<2E-16
Bio1	0.141	1.37	42469	<2E-16
Bio12	0.023	2.14	6348	<2E-16
Joint Contribution	0.608			
Total	0.906			

Reviewer Reports on the First Revision:

Referees' comments:

Referee #1 (Remarks to the Author):

Thank you for addressing my comments and those of the other reviewers.

Referee #2 (Remarks to the Author):

I will start by apologizing for the late review. I don't think I've ever had a harder time making up my mind than with this article. Whether it is actually the beginning of something new, or it represents something really basal. In the end I think that the very fact it has had me think so much is probably a good enough reason for thinking that it is in fact interesting.

Comments on the response letter:

They argue that their gridded climate heatmaps aren't just extensions of a bivariate scatter plot, and I buy that argument.

I was really surprised that log transformation does not change the results more, because I thought they must drastically change the composition and identity of grid cells, and as such species richness and composition. I guess the reason why it does not has to do with many of the precipitation-related variables not being very skewed?

Given that the color gradient might be any color gradient, as they suggest, it might be more revealing to use a color scheme that varies along both axes - it could be a light-dark gradient along pc2, or could be some RGB mix of two color gradients, perhaps?

Now that I've gone more into it I have quite a few comments and questions still.

Before going into the technical details: I don't think I agree with the link to island biogeographical theory. The article argues that reduced gene flow leads to increased speciation in islands, and that this could explain the observation here of more species in more "isolated" climate cells. But on islands, that's not how it works. There is a "radiation zone" where lack of gene flow leads to increased anagenetic speciation, but that does not increase species richness. Outside this, some theory argues of increased speciation rate, but this is not caused by the isolation but by a lower species richness, caused by the isolation. It thus is quite the opposite.

Is it not more likely here that the richness/isolation link is due to more geographical distance among cells with the same climate makes it more likely that they sample from different species pools?

I have a number of concerns with the data analysis.

The text mentions (several places. e.g. x axis of Fig 3b) that isolation is the distance among climate "fragments" (defined in the supp mat as contiguous spatial grid cells in the same climate cell), but

other places it states that isolation is calculated as the mean geodesic distance among all grid cells in the same climate cell. These are very different - which one is it? If it is grid cells, this should be highly correlated to area as a null expectation and hard too see this as an independent measure of isolation. E.g. a very large completely contiguous area would have high isolation. Possibly a null model on fragment size could play a role in addressing this, though mean pairwise distance is a tricky metric when you have clustered grid cells, affected by both number, size and spacing of clusters. I would like to see a more clearly investigation of what aspect of isolation this really catches.

Possibly, does this way of measuring isolation lead to the observed high "isolation" in the tropics - given the simple fact that high geodesic is possible in the tropical band as the earth is widest here? Similarly the upper envelope line of Fig 5b could be shaped by the circumference of the Earth at that latitude maybe?

There is a hard-to-address issue related to the large size of the grid cells, reflecting (perhaps) the uncertainty in species distributions, but which also means that climate isolation gets massively overestimated for many climates that are present in grid cells but just are averaged out.

My biggest concern with the statistical treatment is the use of GAMs. The authors mention that it explains a lot of the variation (measured as R², though I'm not really sure R² is valid for GAMs with Poisson errors), but GAMs are very flexible and good at fitting the data. I'm not a statistician, and I'd suggest checking with one, but normally I would not consider it valid to use GAM fits to reason about the explanatory power of explanatory variables, though that is done here (because parameters are not interpretable as for linear model). The paper mentions that this is due to "non-linearities", but what are these? Are they humps they could be fit with polynomials, and are they not that raises the question what causes these non-linearities if there is a straightforward causal relationship?

Another aspect of the statistical coverage I found strange is that such a large proportion of the variance can be ascribed to either climate or "climate structure", in the sense that R² remains high when one is removed from the model. Yet the authors report very little intercorrelation. How can those factors explain the same variance without being intercorrelated? Is maybe some of the explanatory power derived from the smoothers in the GAM so they are neither climate nor structure? It would be nice to have a statistician evaluate whether this approach to variation partitioning is valid under a GAM.

Line 85 states that these insights should help face the challenge of climate change, I mean, that is really hard to see how, and should at the very least be supported with examples. It is revisited in line 365 but not more convincing here, and it seems a bit superflous.

Minor line-by-line points

The sentence line 16-19 is almost impossible for the reader to understand as it uses concepts that are not introduced yet. Also the dichotomy is false/confusing, as "only the climatic gradients" seems to imply that there are no non-climatic drivers.

line 25: is it really “non-trivial”? Does it not just suggest that these factors are related?

line 26-30 is also quite mysterious to a naive reader, and also the sentence lacks logical cohesion - what does larger regions have to do with the intertwinement of climatic extent and isolation?

line 61: swap large and isolated to match gene flow and heterogeneity for a clearer sentence

line 69-71: why is that a challenge?

77-80: revise sentence for clarity and logical progression

104: “average of the” should possibly be “within-cell average of each of the”

146-147: very hard to understand what this means

229-233: hard to follow

275-277: so is this - not sure it is fully logically consistent

Referee #3 (Remarks to the Author):

Congratulations on the improved manuscript. I evaluated its previous version and have been generally enthusiastic about the presented research. Yet, I made several suggestions (esp. regarding the definition of the climatic space). These have been thoroughly addressed in this round of revisions (e.g. lines 470-474). Consequently, I feel these revisions have further strengthened the presented results and made them accessible to broader audiences, which might have different conceptions of what constitutes the climatic space for their own study organisms.

Overall, the revised study presents a robust and conceptually novel approach toward studying biodiversity patterns, one of the most widely investigated topics in biology. Namely, the authors demonstrate how this line of research might benefit from examining biodiversity patterns in the environmental, in addition to the geographic, space. To this end, they design and implement an original methodology (based on a multidimensional climatic space), which they apply to tetrapod vertebrates (30,000+ species). The resultant conclusions then seem to advance in a compelling way the classic debate on relative importance of climate and geographic area in the emergence of the biodiversity patterns.

In line with my suggestion, the authors have demonstrated in their revised manuscript that their conclusions are robust toward the definition of the climatic space (Tables S3-S10). Namely, they switched NPP and PET, as the measure of environmental productivity, and used the definition of climatic space based on temperature and precipitation (as in the classic Whittaker’s diagram). I

believe that these revisions and additions might be key for future work, as they justify that researchers define the multidimensional space using alternative environmental and climatic variables, which should reflect the biology of their study organism. I also appreciated the argument that using PCA of the environmental variables might be more appropriate than using the environmental variables themselves (last paragraph of page 15 and first paragraph of page 16 in the Cover letter). Perhaps this argument from the Cover letter could also be explicitly mentioned in the text (e.g. as a possible guideline for future researchers in the main text or in the Supplementary Material). Moreover, I have been encouraged to specifically comment on the use of statistics. As in the previous version of the manuscript, the statistical analyses are competently executed, statistical uncertainty, confidence intervals and p-values are properly reported, and results adhere to the rigorous standards in the field.

Considering its novelty on multiple fronts, including the conceptual ideas, methodology and empirical results, I feel this study will be a valuable addition to the literature and might open up new grounds for potentially key research to advance our knowledge of biodiversity, including more efficient ways to safeguard it (e.g. effects of area, isolation, and climate on biodiversity loss).

Author Rebuttals to First Revision:

Revision overview

We are sincerely grateful for considering our manuscript for publication in Nature and for providing us with another opportunity to enhance it. In this second revision, two referees (reviewers 1 and 3) were highly satisfied with our manuscript's updated version, while another (reviewer 2) suggested further improvements concerning the analyses. We have now fully addressed all statistical issues raised by Referee #2, incorporating all suggested additional analyses and making minor amendments to the main text. This valuable feedback has significantly improved our study's clarity and rigor.

Concerning the analyses, we would like to emphasize that Referee #1 thoroughly approved our statistical analysis in the first revision round and expressed strong satisfaction with the second round of revisions. Additionally, Referee #3 has meticulously reviewed and validated all the statistical methods used in our study across both the initial and revised manuscript versions.

Our detailed responses to the referee's comments are provided below:

Point by point responses to reviewers

Referee #1

Thank you for addressing my comments and those of the other reviewers.

We are grateful to the reviewer for dedicating their time and effort to carefully evaluate our revised manuscript, as well as for thoroughly reviewing our detailed responses. Your insightful feedback has been invaluable in improving our work.

Referee #2

"I will start by apologizing for the late review. I don't think I've ever had a harder time making up my mind than with this

article. Whether it is actually the beginning of something new, or it represents something really basal. In the end I think that the very fact it has had me think so much is probably a good enough reason for thinking that it is in fact interesting.

Comments on the response letter:

We appreciate the time and effort the reviewer has dedicated to our revised manuscript. The continued constructive feedback has allowed us to make sure our approach is sound and our arguments clear.

They argue that their gridded climate heatmaps aren't just extensions of a bivariate scatter plot, and I buy that argument.

I was really surprised that log transformation does not change the results more, because I thought they must drastically change the composition and identity of grid cells, and as such species richness and composition. I guess the reason why it does not has to do with many of the precipitation-related variables not being very skewed?"

As stated earlier (*Supplementary Material, Tables S15 to S18 and Fig S10*), implementing a log transformation on the variables prior to conducting the PCA has minimal impact on the spatial patterns of the resulting PCA scores. Whether the precipitation variables are transformed or not, the correlation between the PCA scores remains remarkably high. This suggests that potential skewness in these variables does not substantially alter the spatial patterns detected in the PCA scores when these variables are included alongside several others.

Referee #2

Given that the color gradient might be any color gradient, as they suggest, it might be more revealing to use a color scheme that varies along both axes - it could be a light-dark gradient along pc2, or could be some RGB mix of two color gradients, perhaps?

We're grateful for the feedback regarding the color selection in our initial figure. After exploring various options, we've now chosen a color scheme that more clearly illustrates transitions between climate cells. **As a result, we've updated Figure 1 accordingly.** Please find below a visualization that incorporates the reviewer's suggestion.

We also understand that finer details may not be easily discernible when the world map is represented in a small image, as it is in Figure 1. To address this concern, **we've included a separate larger map as supplementary material (Fig S14)** that can be see bellow:

Referee #2

Now that I've gone more into it I have quite a few comments and questions still.

Before going into the technical details: I don't think I agree with the link to island biogeographical theory. The article argues that reduced gene flow leads to increased speciation in

islands, and that this could explain the observation here of more species in more “isolated” climate cells. But on islands, that’s not how it works. There is a “radiation zone” where lack of gene flow leads to increased anagenetic speciation, but that does not increase species richness. Outside this, some theory argues of increased speciation rate, but this is not caused by the isolation but by a lower species richness, caused by the isolation. It thus is quite the opposite.

Is it not more likely here that the richness/isolation link is due to more geographical distance among cells with the same climate makes it more likely that they sample from different species pools?

We appreciate your valuable feedback. We concur that our initial analogy, linking speciation rate with island isolation, might be confusing. Consequently, **we've updated the manuscript (Line 62) to remove this comparison.**

Our key point is that larger richness and turnover within isolated climates could result from two factors: geographical isolation facilitating speciation, and unique biotas evolving independently within similar, but isolated, climate conditions. This has been stressed in our manuscript (**Lines 231-239**). The boost in species richness and turnover with climate isolation likely stems from a mix of isolation and the fact that the more isolated a climate is, the more

likely it is to sample from different species pools due to differing evolutionary histories.

While discussing potential mechanisms, we uphold that a connection exists between the isolation of climate conditions and biodiversity diversification. The isolation of certain climates can result in isolated populations, thus influencing diversification rates. This idea has been backed by mechanistic simulation models that effectively replicate empirical biodiversity patterns (Rangel et al. 2018).

Rangel, T. F. *et al.* Modeling the ecology and evolution of biodiversity: Biogeographical cradles, museums, and graves. *Science* **361**, eaar5452 (2018).

Referee #2

I have a number of concerns with the data analysis.

The text mentions (several places. e.g. x axis of Fig 3b) that isolation is the distance among climate “fragments” (defined in the supp mat as contiguous spatial grid cells in the same climate cell), but other places it states that isolation is calculated as the mean geodesic distance among all grid cells in the same climate cell. These are very different - which one is

it? If it is grid cells, this should be highly correlated to area as a null expectation and hard too see this as an independent measure of isolation. E.g. a very large completely contiguous area would have high isolation. Possibly a null model on fragment size could play a role in addressing this, though mean pairwise distance is a tricky metric when you have clustered grid cells, affected by both number, size and spacing of clusters. I would like to see a more clearly investigation of what aspect of isolation this really catches. Possibly, does this way of measuring isolation lead to the observed high "isolation" in the tropics - given the simple fact that high geodesic is possible in the tropical band as the earth is widest here? Similarly the upper envelope line of Fig 5b could be shaped by the circumference of the Earth at that latitude maybe?

We appreciate the reviewer's concern about a consistent and clear use of terms. In the interest of clarity, we want to underscore that our main analysis utilizes a measure of **distance between climate fragments**. We believe that the confusion was caused by a mistaken reference to "the average distance among all geographical cells that occur within a climate cell", which, in the last version of the manuscript, was in the caption of Figure 1. The mistake has been corrected. **We consistently edited the main text to remove any potential confusion with other metrics of isolation (Line 106, Lines 129-131, Line 210, Line 245, Lines 545-559).**

However, in the methodology of the paper we do indeed mention that other measures of climate isolation were considered. The choice outlined in the study's methodology directly replies to the reviewer's concerns about the association with climate area (**Lines 545-559**) : *"To better represent climate isolation, we used **three alternative metrics, but because some of them are strongly correlated with climate area and also correlated to each other we chose one of the three for the final analysis.** We measured climate isolation as (i) the mean geodesic distance among all geographical cells that occur within a climate cell. Alternatively, we considered that some geographical cells that occur within a climate cell form fragments of climates (geographical cells connected to each other that occur within a single climate). Therefore, the size of climate fragments could affect the isolation measures. To control for this effect we (ii) measured the average distance between climate fragments that occur within a climate cell and the (iii) number of climate fragments within a climate cell. **The total count of fragments within a climate cell has a strong link with the overall climate area (with a Pearson's correlation of 0.94). Similarly, the average distance between all geographical cells within a climate cell is also strongly related to the average distance between climate fragments (Pearson's correlation of 0.88). In a process where we calculate the average distance between all geographical cells within a climate cell, larger climate fragments may have a greater impact on the average because they provide more points of comparison. Therefore, in our final models, we chose to use the average distance between climate fragments within a climate cell.**"* Finally, the correlation between the average distance between climate fragments and climate area is low (*Pearson's correlation of 0.11*) which allows us to use this measure with climate area in the same model.

We appreciate the reviewer's concern regarding the potential artifactual correlation between the total area of a latitudinal belt (such as the tropics) and climate isolation, as

defined by the mean distance between grid cells. We agree that, if climates were randomly distributed across the globe, climate isolation would indeed strongly correlate with latitude. However, we calculated climate isolation based on the distance between climate fragments, a measure less influenced by total area availability. Furthermore, climates are not randomly distributed - they typically mirror each other across hemispheres. Therefore, contrary to the reviewer's null expectation, biogeographic understanding generally acknowledges a striking pattern of climate isolation: two broad climatic zones in high latitudes (northern and southern polar and temperate zones) are disjointed, divided by a large, contiguous tropical zone (**Lines 240-243**). In our research, we directly assess the influence of area and isolation of individual climates on biodiversity patterns, in addition to the direct effects of climatic conditions. Notably, our analysis reveals that tropical climates are more fragmented than commonly assumed.

Referee #2:

My biggest concern with the statistical treatment is the use of GAMs. The authors mention that it explains a lot of the variation (measured as R², though I'm not really sure R² is valid for GAMs with Poisson errors), but GAMs are very flexible and good at fitting the data. I'm not a statistician, and I'd suggest checking with one, but normally I would not consider it valid to use GAM fits to reason about the explanatory power of explanatory variables, though that is done here (because parameters are not interpretable as for linear model). The paper mentions that this is due to "non-linearities", but what are these? Are they humps they could be fit with polynomials, and are they not that raises the question what causes these non-linearities if there is a straightforward causal relationship?

The reviewer's insightful comment has been taken into consideration, leading to the implementation of additional analyses to verify the robustness of the results. **Detailed responses to the points raised by the reviewer are provided below.** In summary: (1) the reviewer's query about the use of R-squared as a measure of fit for nonlinear models with non-normally distributed errors has been addressed. The results are now presented with percentage of null deviance in addition to adjusted R² (**Lines 139-147**, Supplementary Tables). The results from both measures align closely, and the choice between the two does not alter the conclusions or interpretations of the study. (2) In the previous version, the term "variance" was mistakenly used when reporting explanatory power, although "deviance" was the measure used to evaluate goodness-of-fit. This error might have prompted the reviewer to question the use of standard R-Squares (as in OLS), which is indeed problematic in GAM. However, the use of percentage of null deviance in GAM is standard and well accepted. The text has been corrected to accurately reflect the measures used, which are the percentage of null deviance and adjusted R² (**Lines 139-166**), all appropriate for GAM with Poisson errors.

(3) A simpler linear model explains approximately 76% of observed data, which is sufficient to support the main conclusion. However, the residuals from this model indicate a violation of the linearity assumption. (4) The non-linearities present in the residuals of the linear model can be accommodated with a polynomial model of degree four. This model results in a modest 14% increase in goodness-of-fit compared to the linear model. (5) the polynomial regression yields the same results to the GAM. (6) Although no clear theoretical explanation for the observed non-linearity is currently available, the use of the non-linear model was a necessary statistical adjustment given the violation of key assumptions by the linear model. Nonlinearities could have various causes, as detailed in the response below. However, we believe that non linearity in the residuals belong to the realm of epistemic uncertainty (could potentially be explained with additional knowledge), as opposed to aleatoric uncertainty (noise or randomness).

Detailed response to the comment above

[...]The authors mention that it explains a lot of the variation (measured as R2, though I'm not really sure R2 is valid for GAMs with Poisson errors).

Goodness of fit measure:

The reviewer highlights a valid concern regarding the use of standard R-squared in non-linear models with non-normally distributed errors. R-squared is based on the proportion of variance explained, assuming normally distributed errors, such as in standard OLS models. However, in our study we used percentage of null deviance, which is a measure of goodness-of-fit suitable for non-linear models with non-normally distributed errors. Deviance quantifies the discrepancy between observed data and model predictions, with lower deviance value indicating a better fit. The proportion of null deviance explained compares the null deviance (fit for a simple model with only an intercept and, if present, an offset) to the residual deviance (the fitted model's deviance). This proportion shows the improvement in fit achieved by the more complex model over the null model. As suggested by Wood (2017), the proportion of null deviance may be more appropriate for non-normal errors, as it accounts for the specific distribution and link function of Generalized Linear Models (GLM) or Generalized Additive Models (GAMs), which are extensions of GLMs. In addition, to assess the importance of each predictor, as measured by the contribution to the total explanatory power, we “*fit alternative models without each predictor and computed a reduction in deviance*” (quote from the main text, **Lines 593-599**). Detailed procedure to compute the importance of each predictor is detailed in a further response to a specific comment of the reviewer on this topic.

We acknowledge that, in the last version of our manuscript, we mistakenly referred to “*variance*”, instead of the proportion of null deviance, in certain sections of the main text and supplementary tables when discussing model fit, which may have caused the misunderstanding by leading the reviewer to believe that we were using standard R-squared. We have now updated the manuscript to report both the proportion of null deviance explained and adjusted R-squared (**Lines 139-147** and Supplementary Tables). There is a small (approximately 2%) difference between these two measures of goodness-of-fit, which does not affect, in any way, the interpretation and conclusion of the manuscript.

In conclusion, we hope to have not only answered the reviewer about the validity of our measure of goodness-of-fit for GAMs with Poisson errors, but also showed that our results are robust to different goodness-of-fit measures. **We have addressed the reviewer's concern by (1) reporting both proportion of null deviance and adjusted R-squared, which do not differ substantially, and (2) correcting the manuscript to properly refer to the proportion of null deviance, instead of variance.**

Wood, S. N. *Generalized Additive Models: An Introduction with R, Second Edition*. (Chapman and Hall/CRC, 2017). doi:[10.1201/9781315370279](https://doi.org/10.1201/9781315370279).

[...]but GAMs are very flexible and good at fitting the data. I'm not a statistician, and I'd suggest checking with one, but normally I would not consider it valid to use GAM fits to reason about the explanatory power of explanatory variables, though that is done here (because parameters are not interpretable as for linear model). The paper mentions that this is due to "non-linearities" [...] Are they humps that could be fit with polynomials? [...]

Flexibility of GAMs and the potential use of polynomial regression

The reviewer brings up an important question regarding our choice to explore non-linearity in our study using Generalized Additive Models (GAMs). The reviewer also asked if polynomial regression would be sufficient to account for the non-linearities. We have rerun all the analysis using polynomial regressions (Tables S19 to S23) to compare the results against those from GAMs, and, as shown below, we found no difference between the two models.

It is crucial to assess the explanatory gain when moving from a linear model to a non-linear model. Since our response variable is richness, a count data represented by an integer variable equal to or greater than 1, a Poisson link is a natural choice, leading us to a polynomial GLM. Thus, we defined a GLM with richness as the response variable and geography of climate (i.e., area and isolation) and climate itself (i.e., PC1 and PC2) as predictors. **For the patterns of bird species richness the linear GLM model explains 76% (0.763), as measured by null deviance, and 75% (0.756) as measured by R-squared (MacFadden's R^2).** Other groups of species show similar results (Table S19). **We highlight that a simple linear model explains species richness patterns at a level that could justify the main conclusion of our study.**

However, as shown below (Figure R1 extracted from Fig S15), the Residuals vs Fitted plot shows non-linearities in the adjusted linear GLM model, therefore violating the assumptions of the linear model and deeming estimated parameters uninterpretable.

Fig R1. Residual vs Fitted Plot for the Linear Model. This plot illustrates the residuals (the differences between the observed and predicted values) on the y-axis against the fitted values (predicted values) on the x-axis for the linear Generalized Linear Model (GLM). The aim of this plot is to diagnose the model's fit and to identify potential non-linearity and heteroscedasticity in the data. Ideally, if the model is a good fit and the assumptions of linearity and homoscedasticity hold, the residuals should be randomly scattered around the horizontal zero line across the range of fitted values. Systematic deviations from this line indicate issues with model fit and violations of model assumptions.

As wondered by the reviewer, we tried to address non-linearity using a polynomial regression, in which the *polynomials are made orthogonal*, so that collinearity issues are not introduced into the model.

As with any non-linear model, the challenge lies in defining the flexibility of the model to the data. On one hand, if the model is too rigid it may not be sufficient to account for the non-linearities. On the other hand, if the model is too flexible it may overfit, therefore accounting for noise in the data. In polynomial regression the flexibility of the model is measured as polynomial degree.

Here we compared the linear GLM against non-linear polynomial GLMs of four different degrees of flexibility, ranging from two to five (see Table R1 as an example for birds). From the additional analysis using polynomial GLM we conclude that:

- (1) the 4th degree polynomial GLM provides the smallest deviance (Table R1, column “deviance”) and successfully addresses the non-linearity issue (Fig. R2)
- (2) the four-degree polynomial GLM has goodness-of-fit of approximately 90% (0.908 measured by null deviance, or 0.902 measured by R-Squared, Table S19), which is equivalent to the goodness-of-fit of GAM with $k = 4$ (0.90 measured by null deviance, or 0.92 measured by adjusted R^2).

(3) the gain in goodness-of-fit from a linear to a four-degree polynomial model is 14%, which is equivalent to the gain in goodness-of-fit provided by the GAM with $k = 4$ compared to the linear model.

(4) because a 4th degree polynomial GLM does not seem to be overfitting the data, and because the goodness-of-fit of polynomial GLM and GAM are equivalent, we can conclude that GAM is also not overfitting the data.

Table R1. Comparison of Deviance Reduction for Linear and Polynomial Generalized Linear Models. This table presents the results of an Analysis of Variance (ANOVA) test performed on a set of Generalized Linear Models (GLMs) with varying degrees of polynomial terms, as well as the gain in explanatory power with different degrees of non-linear flexibility. The GLMs range from a linear model (no polynomial terms) to polynomial models of degrees two through five. The purpose of this ANOVA is to compare the goodness-of-fit of these models to the data, as indicated by the reduction in deviance from one model to the next. The deviance, a measure of the discrepancy between the observed data and the model's predictions, is given for each model, showing how well each model fits the data. Lower deviance values indicate a better fit. Please note that the linear model does not have a deviance value as it serves as the reference model for comparison with the polynomial models. Values for all groups are presented on Table S19. Column "McFadden's R^2 " indicates the goodness-of-fit of each GLM as measured by McFadden's pseudo R^2 , and column "Delta R^2 " indicates the gain in goodness-of-fit by the non-linear models when compared to the linear model.

Models	Residual Deviance	Degrees of Freedom	Deviance	McFadden's R^2	Delta R^2 (Polynomial - Linear)
Linear	54674			0.758	
Polynomial 2nd degree	24799	4	29875.1	0.879	0.121
Polynomial 3rd degree	20791	4	4008.1	0.898	0.140
Polynomial 4th degree	19852	4	939.5	0.902	0.144
Polynomial 5th degree	18337	4	1514.4	0.909	0.151

Fig R2. Residual vs Fitted Plot for the 4th degree Polynomial Model. This plot displays the residuals against the fitted values for the selected Polynomial Generalized Linear Model (GLM). The absence of systematic deviations from the horizontal zero line suggests a good model fit, indicating the polynomial model effectively accounts for the non-linearity in the data.

In fact, overfitting concerns is what led us to use GAMs rather than polynomial GLM. Since the first version of our study, we have been explicit about the limitations we imposed on the non-linear flexibility of GAMs: *"we used Poisson Generalized Additive Models (GAMs) with penalized smooth functions conducted through Generalized Cross Validation (GCV)^{43,44}. GCV trades-off curve complexity against goodness of fit to avoid complex over-fitted estimates. Basis dimension choices for smooth terms were set to four ($k = 4$) since patterns in our residuals are not better explained by higher dimensions of k ."*

We opted to use GAMs in our main text because GAMs are extensions of GLMs, being a safer and simpler option over polynomial regressions, which were originally designed for linear regression with normally distributed errors. In fact, GAMs offer two main advantages over polynomial regression: (1) Reduced overfitting: GAMs use inherently regularized smoothing functions, helping to reduce overfitting, a common issue in polynomial regression. GAMs control smoothness through smoothing parameters, providing a balance between model fit and complexity that prevents overfitting. (2) Robustness to outliers: GAMs are more robust to outliers than polynomial regression, as the smoothing functions used in GAMs are less influenced by extreme data points. Polynomial regression models can be highly sensitive to outliers, leading to misleading results and poor generalization performance.

Finally, the reviewer rightly points out that interpreting non-linear models can be more challenging than that of linear models. In our paper our primary goal is to describe the strength and form of the relationships, rather than providing parameter estimates for predictions. Describing strength and form of statistical relationships in observed data (Fig. 2), followed by a detailed theoretical discussion of potential mechanisms underlying the

observed relationships, is standard scientific practice in macroecology, which is a quintessential historical science that does not allow direct experimental manipulation.

The procedures above are now mentioned in the main text (**Lines 566-592**) and detailed in the supplementary material.

[...]The paper mentions that this is due to “non-linearities”, but what are these? [...] what causes these non-linearities if there is a straightforward causal relationship? [...]

The underlying causes of non-linearities

The reviewer inquires about the nature of the non-linearities observed in our study. This is indeed a complex question and widely debated in statistics and inferential philosophy. Unfortunately, we have no definite answer, but we believe that non-linearities observed in our study belong to the realm of epistemic uncertainty, that is, could eventually be explained with additional knowledge. However, as wildly speculative as the conjectures about drivers of non-linearity are, we believe they fall beyond the scope of our manuscript, and should be investigated in future studies, under the light of ecological and evolutionary theory yet to be developed.

Importantly, as described above, **we highlight that a simple linear model explains around 76% of the observed species richness patterns. Thus, although there is unexplained non-linear information, the simple linear model is sufficient to support our conclusions about the role of the geography of climate and climate itself on biodiversity patterns. In addition, our use of a non-linear model was strictly driven by the statistical necessity to account for non-linearity in the residuals, and to evaluate if a non-linear model provided different results than a linear model. Finally, our interpretation of the fit of our model to empirical data does not rely exclusively on a non-linear model.**

Still, we structure our (rather speculative) answer to the non-linearity problem around the three theoretical causes of non-linearity: **(a)** genuine causal non-linear relationships between predictors and the response variable, **(b)** non-stationarity in the relationship between the response and predictor, or **(c)** model misspecification by inadequate or missing predictors, and other statistical causes.

It is not difficult to conceive that many biological processes may be truly non-linear. In fact, they may be more common than usually acknowledged in macroecology. For example, it has been hypothesized that species richness drives speciation rates (Emerson and Kolm 2005), therefore leading to non-linear patterns. Allometric scaling, population dynamics and island biogeography are additional examples of non-linear patterns that can be studied under existing theory. However, at this point we have no theory to suggest ecological or evolutionary processes driving non-linearities in species richness patterns.

A causal process is called non-stationary if the relative importance of different factors driving a pattern shifts over time or space. Non-stationary processes may drive non-linear patterns in the data, especially if the residuals of models erroneously assume stationarity. In

macroecology it is still uncommon to relax the stationarity assumption even in standard biodiversity analysis in geographic or phylogenetic spaces, even though non-stationary models, such as Geographically Weighted Regression, have been available for a while (eg., Barreto et al. 2021). We believe that it is theoretically premature to associate the observed non-linearities present in the residuals of our study to non-stationary causal processes, as these non-linearities are possibly due to true non-linear causal processes or model misspecification. Still, as we establish the theoretical framework for macroecological analysis in climate space, future studies may find compelling associations between observed non-linearities with non-stationary processes, such as the imprint of rare evolutionary events.

If a model is lacking an important causal factor, or if the pattern is driven by the interaction among factors that is not specified in the model, then non-linearity in the residuals may emerge due to model misspecification. As indicated by the explanatory power of a simple linear model (approximately 76%), the commonness and spatial distribution of climate conditions capture most of the effect of the geography of climates on biodiversity patterns. Additionally, defining climate conditions using orthogonal axes derived from a dozen climatic variables that impose limits to species distributions is sufficient to represent the multivariate effects of climate on biodiversity. Still, our model is missing several potential interactions among these factors, which could be the drivers of the observed non-linearities in the residuals. Most importantly, historical contingencies and the evolutionary history of the clades are also missing from the model, as they are inherently difficult to estimate and to be encapsulated in a simple predictor variable. These, and many other yet to be discovered ecological and evolutionary factors, could be the causes of the observed non-linearities in the residuals.

Barreto, E. *et al.* Spatial variation in direct and indirect effects of climate and productivity on species richness of terrestrial tetrapods. *Glob. Ecol. Biogeogr.* **30**, 1899–1908 (2021).

Emerson, B. C. & Kolm, N. Species diversity can drive speciation. *Nature* **434**, 1015–1017 (2005).

Referee #2

Another aspect of the statistical coverage I found strange is that such a large proportion of the variance can be ascribed to either climate or “climate structure”, in the sense that R² remains high when one is removed from the model. Yet the authors report very little intercorrelation. How can those factors explain the same variance without being intercorrelated? Is maybe some of the explanatory power derived from the smoothers in the GAM so they are neither

climate nor structure? It would be nice to have a statistician evaluate whether this approach to variation partitioning is valid under a GAM.

To respond to the reviewer's concern we (1) explained in detail the reason why there can be around 60% shared explanatory power among predictors while the intercorrelation among pairs of predictors is small, and (2) we revised the analysis to further partition the explanatory power of the model (*Lines 158-166, Fig 4*).

Partitioning of explanatory power and sources of shared explanation

The explanatory power that can be ascribed uniquely to an individual predictor can be calculated as the reduction in goodness-of-fit (e.g. proportion of null deviance or adjusted R^2) when the predictor is removed from the full model. For example, our full model has a proportion of null deviance of approximately 90%, whereas the model that does not include climate area has a proportion of null deviance of approximately 80%. Thus, the isolated contribution of climate area, as measured by the proportion of null deviance (or adjusted R^2), is approximately 10%. The non-overlapping and unique explanatory power provided by all the predictors is the sum of the independent contribution of each predictor, which is approximately 30% in our study. In other words, 30% is the amount of explanation that can be ascribed uniquely and individually to one of the predictors.

The difference between the explanatory power of the full model (90%) and the combined unique individual contribution of all predictors (30%) is the total joint contribution (60%).

The reviewer wonders "*How can those factors explain the same variance without being intercorrelated?*"

Although there is very little pairwise intercorrelation between each pair of predictors, and VIF indicates that the model does not have multicollinearity problems, the **total joint contribution of all predictors sums up to 60% because it is given not only by the sum of pairwise intercorrelation, but also by the total redundant explanatory power of any non-singleton set of predictors (pairs, triples and quadruple sets of predictors)**. In other words, the 60% includes not only the redundant explanation between Climate Area and Climate Isolation, PC1 and PC2 (pairs), but also the redundant explanation between Climate Area, Climate Isolation and PC1, Climate Isolation, PC1 and PC2 (triplets) and between Climate Area, Climate Isolation, PC1 and PC2 (quadruple).

It is true that there would be no shared explained variance if all the four predictors were perfectly orthogonal to one another. However, the little intercorrelation between predictors sums up to 60% when not only the pairwise intercorrelation is accounted for, but also the intercorrelation between all the additional 4 sets of triplets and the set of quadruplet predictors.

Expanding the partition of explanatory power

In the revised version of the manuscript we further partitioned the explanatory power of the model by breaking down the “joint contribution” into three categories of shared explanation: joint explanation within geography of climate (shared explanation between predictors area and isolation), joint explanation within climate itself (shared explanation between predictors PC1 and PC2), and shared explanation across categories geography of climate and climate itself (see new Fig. 4).

We now show that, at the grid resolution 20 equal intervals, the shared explanatory power within geography of climate is approximately 12%, while within climate itself is around 1 to 2% (at finer climate space resolutions it converges to zero, e.g. Tables S4).

The remaining shared explanation, of approximately 45%, is due to four pairs of predictors across different categories (e.g. area and PC1, isolation and PC2), four triplets (e.g. area, PC1 and PC2) and one quadruplet (area, isolation, PC1 and PC2).

Although the remaining shared explanation could be further partitioned among the nine remaining possible combinations of sets of predictors, we decided to maintain the remaining shared explanation lumped together under the category of “joint Geography of Climate and Climate Itself”, as the additional partitions would not contribute to the interpretation of results.

The reviewer has raised the possibility that our variation partitioning strategy could not be applicable to GAM. As suggested by the reviewer, we also fitted polynomial GLMs. Importantly, because the percentage of null deviance is equivalent between GAM and polynomials, by definition the partitions are also equivalent (Tables S20 to S23). **The statistical requirement for partitioning of explanatory power is (1) a statistically valid measure of goodness-of-fit for the given model (2) that ranges within a bounded domain (such as percentage of null deviance, R^2 , unlike SSE or likelihood).** Thus, we are confident that the statistical methodology that we employed to partition explanatory power is robust and was explicitly validated by reviewer #3 and implicitly by reviewer #1.

Referee #2:

Line 85 states that these insights should help face the challenge of climate change, I mean, that is really hard to see how, and should at the very least be supported with examples. It is revisited in line 365 but not more convincing here, and it seems a bit superflous.

We appreciate the reviewer's feedback and have accordingly amplified the discussion around the spatial dynamics of climate area, isolation, and protection levels of climates under climate change (**Lines 312-354**).

Referee #2:

Minor line-by-line points

The sentence line 16-19 is almost impossible for the reader to understand as it uses concepts that are not introduced yet. Also the dichotomy is false/confusing, as “only the climatic gradients” seems to imply that there are no non-climatic drivers.

We agree with the reviewer and re-wrote the abstract of the manuscript to be more accessible to the general reader (**Lines 16 to 32**).

line 25: is it really “non-trivial”? Does it not just suggest that these factors are related?

We already gave a detailed response to the reviewer about the joint contribution of the geography of climate and climate itself, which is what is mentioned by the reviewer here. In any case, the abstract has now been edited based on the comment above (**Lines 16-32**).

line 26-30 is also quite mysterious to a naive reader, and also the sentence lacks logical cohesion - what does larger regions have to do with the intertwinement of climatic extent and isolation?

We agree with the reviewer about the logical cohesion of the sentence which has been properly edited (**Lines 27-32**).

line 61: swap large and isolated to match gene flow and heterogeneity for a clearer sentence

We incorporated the suggestion (**Line 65**)

line 69-71: why is that a challenge?

We properly edited this sentence to explicitly say why it is challenging to assess the impact of the geography of climate (**Lines 70-74**).

77-80: revise sentence for clarity and logical progression

We divided this sentence in two for clarity and logical progression (**Lines 81-84**).

104: "average of the" should possibly be "within-cell average of each of the"

Corrected (**Line 108**)

146-147: very hard to understand what this means

This sentence was revised to increase its clarity (**Lines 151-153**).

229-233: hard to follow

This sentence was revised to increase its clarity (**Lines 231-235**).

275-277: so is this - not sure it is fully logically consistent

This sentence was revised to increase its clarity (**Lines 281-284**).

Referee #3:

Congratulations on the improved manuscript. I evaluated its previous version and have been generally enthusiastic about the presented research. Yet, I made several suggestions (esp. regarding the definition of the climatic space). These have been thoroughly addressed in this round of revisions (e.g. lines 470-474). Consequently, I feel these revisions have further strengthened the presented results and made them accessible to broader audiences, which might have different conceptions of what constitutes the climatic space for their own study organisms.

Overall, the revised study presents a robust and conceptually novel approach toward studying biodiversity patterns, one of the most widely investigated topics in biology. Namely, the authors demonstrate how this line of research might benefit from examining biodiversity patterns in the environmental, in addition to the geographic, space. To this end, they design and implement an original methodology (based on a multidimensional climatic space), which they apply to tetrapod vertebrates (30,000+ species). The resultant conclusions then seem to advance in a compelling way the classic debate on relative importance of climate and geographic area in the emergence of the biodiversity patterns.

In line with my suggestion, the authors have demonstrated in their revised manuscript that their conclusions are robust

toward the definition of the climatic space (Tables S3-S10). Namely, they switched NPP and PET, as the measure of environmental productivity, and used the definition of climatic space based on temperature and precipitation (as in the classic Whittaker's diagram). I believe that these revisions and additions might be key for future work, as they justify that researchers define the multidimensional space using alternative environmental and climatic variables, which should reflect the biology of their study organism. I also appreciated the argument that using PCA of the environmental variables might be more appropriate than using the environmental variables themselves (last paragraph of page 15 and first paragraph of page 16 in the Cover letter). Perhaps this argument from the Cover letter could also be explicitly mentioned in the text (e.g. as a possible guideline for future researchers in the main text or in the Supplementary Material). Moreover, I have been encouraged to specifically comment on the use of statistics. As in the previous version of the manuscript, the statistical analyses are competently executed, statistical uncertainty, confidence intervals and p-values are properly reported, and results adhere to the rigorous standards in the field.

Considering its novelty on multiple fronts, including the conceptual ideas, methodology and empirical results, I feel this study will be a valuable addition to the literature and might open up new grounds for potentially key research to advance our knowledge of biodiversity, including more efficient ways to safeguard it (e.g. effects of area, isolation, and climate on biodiversity loss).

We are grateful for the referee's commitment to invest time in our manuscript and for their thoughtful and encouraging comments on our work throughout both the previous and current revisions. We also appreciate the valuable insights regarding the use of statistical methods in our study. In response to the referee's suggestion, we have now included a rationale for using PCA over the environmental variables themselves in the current version of the manuscript, which may serve as a useful guideline for future research (**Lines 513-512**).

Reviewer Reports on the Second Revision:

Referees' comments:

Referee #2 (Remarks to the Author):

I'd like to thank the authors for engaging thoroughly and constructively with my comments. I remain not wholly convinced about the main point of the paper - that isolation of climate in gridded climate space should be a key driver of species richness - but I do think the idea and presentation here are interesting and should spark fruitful discussion.

I have just a few very minor comments left:

1. The methods don't list anything about whether the climate variables were standardized prior to or during the performance of the PCA. I assume they were, as PCA on unstandardized variables is not valid when the units of the input axes differ, but it's not listed. Please check and include.
2. The procedure involves multiple steps of applying a grid. The first step is geographical aggregation of climate variables. It says that climate was extracted at 110 x 110 km, but when was the aggregation of the input data done (prior or after the PCA?) and how were they aggregated (centroid, mean?)?

Referee #3 (Remarks to the Author):

Thank you for thoughtfully addressing my concerns from the previous rounds. I am satisfied with the revisions made, and I am convinced that this manuscript will make a valuable contribution to the field. It advances current research on multiple fronts. One notable aspect is the introduction of a new methodological strategy, namely the examination of diversity gradients in both the geographic and climatic space. The authors have applied this strategy in a compelling manner to tackle one of the classic and long-standing questions in biology, which pertains to the causes of biodiversity gradients. The results of this study present novel and non-trivial conclusions, revealing that biodiversity gradients are influenced by both climate and climatic isolation. This finding holds meaningful implications for various research areas, including species-area relationships, island biogeography, and conservation biology. As a result, I believe this study will be of interest to a broad readership.

After the previous rounds of detailed revisions, I do not have any substantial recommendations or suggestions. The statistical analyses are competently executed, as previously, statistical uncertainty, confidence intervals and p-values are properly reported, and results adhere to the standards in the field. In my assessment, the manuscript is well-developed and the conclusions compelling. I have

been asked to specifically comment on some of the technical and statistical aspects raised by Reviewer #2. Hopefully, my feedback here will be of use to the authors and will help to address some of the concerns of the reviewer.

(1) Distances among grid cells. The authors use two approaches, where they measure distance between climate fragments and the pair-wise distances between grid cells. Both approaches seem to support the same qualitative results. Perhaps the authors could suggest that future studies use both approaches and compare the results, given that neither approach seems inherently better than the other. Reviewer #2 highlights the possibility that isolation tends to be greater in the tropics. I agree that the longitudinal distances are greater near the equator, however, the authors use the geodesic distance (shortest distance between two points on the surface of the Earth). Technically, the geodesic distances are inherently equivalent around the globe, regardless of where the two points are positioned on Earth. When comparing points between regions with tropical climates, we deliberately choose regions where the biotas have similar climatic adaptations, hence inserting biological knowledge into the analysis. Consequently, the distances will be greater in the tropics, but I would not interpret this as a methodological issue. The fact that regions with tropical climates are large and geographically spread out is arguably one of the reasons for high tropical diversity, which has biological causes (e.g. area for speciation and dispersal barriers leading to allopatry tend to be greater in the tropics). I would caution against statistically removing these effects, given that they might reflect the biological mechanisms behind the formation of diversity gradients, rather than being an artifact of how the distances are technically measured.

(2) GAM models. I also agree with Reviewer #2 that the R^2 values (% of explained variance) are hard to interpret when using GAM with the Poisson error structure. One way to circumvent the issue would be to use the fitted GAM models for predicting the observed values, then correlating the predicted values (predicted diversity, extracted from GAM) with the observed values (empirical diversity). The resultant correlation (Pearson's correlation coefficient), when squared, corresponds to R^2 . These values are easily interpretable and well-known from the linear regression, while the approach is widely used and has good precedent in biodiversity studies (e.g. Jetz and Fine 2012, Plos Biology, Figure 2). The authors could compare these R^2 values with GAM results on % of explained null deviance. I also agree that one would expect the effects of climate and climate structure to be correlated, given the wide overlap between their effects (Figure 4 in the main text). Perhaps this could be reconciled by explicitly referring to partial correlations (e.g. what climate explains) and marginal correlations (e.g. what climate explains after accounting for the effects of climate structure). This could explain why the marginal effects are uncorrelated, even though the effects of climate and climate structure largely overlap. These technical issues seem to be mostly minor, in my assessment, and easy to address, using supplementary analyses.

Author Rebuttals to Second Revision:

We are sincerely grateful for the opportunity to publish our manuscript in Nature. Your willingness to consider our work has been truly appreciated.

We have diligently addressed the final technical comments offered by the reviewers. We believe these revisions have greatly improved both the quality and clarity of our manuscript, and we're very pleased with the final result. We wholeheartedly support and are eager to participate in the process of transparent peer review, which includes publishing reviewers' comments and our response letters.

Thank you once again for your attention and consideration throughout this process.

Technical additions based on reviewers comments:

We've addressed the methodological refinements proposed by Referee #2 and Referee #3, carefully ensuring word count remains unchanged. We've also incorporated the new goodness-of-fit measure recommended by Referee #3, which aligns identically with the previously displayed adjusted R^2 results. Furthermore, our findings already account for the influence of each predictor while controlling for other predictors, eliminating the necessity for partial correlations as proposed by Referee #3. Please find below detailed responses to all the comments.

Editorial Changes to comply with Nature Standards:

We've diligently implemented all editorial suggestions which encompass adding references to the abstract, generating a guide for the supplementary material, and transitioning some display items from the supplementary material to the extended data section. Additionally, we have made modifications to both the sizes of items within the figures and the overall dimensions of the figures themselves.

Reviewers' comments:

Referee #2:

I'd like to thank the authors for engaging thoroughly and constructively with my comments. I remain not wholly convinced about the main point of the paper - that isolation of climate in gridded climate space should be a key driver of species richness - but I do think the idea and presentation here are interesting and should spark fruitful discussion.

We extend our sincere appreciation to the referee for the consistent dedication of time and expertise to our manuscript throughout the various rounds of revisions. The recognition that our study will likely ignite stimulating discussions within the field is indeed gratifying.

As to the subject of isolation, our analysis has been undertaken in the most robust manner possible. Therefore, the resultant findings are grounded in the discernible patterns exhibited by distinct groups of terrestrial vertebrates on a global scale. It is noteworthy that Referee #3 has articulated some aspects of our research in an even more comprehensive manner than our previous revision. These insights include: “*Reviewer #2 highlights the possibility that isolation tends to be greater in the tropics. I agree that the longitudinal distances are greater near the equator, however, the authors use the geodesic distance (shortest distance between two points on the surface of the Earth). Technically, the geodesic distances are inherently equivalent around the globe, regardless of where the two points are positioned on Earth. When comparing points between regions with tropical climates, we deliberately choose regions where the biotas have similar climatic adaptations, hence inserting biological knowledge into the analysis. Consequently, the distances will be greater in the tropics, but I would not interpret this as a methodological issue. The fact that regions with tropical climates are large and geographically spread out is arguably one of the reasons for high tropical diversity, which has biological causes (e.g. area for speciation and dispersal barriers leading to allopatry tend to be greater in the tropics). I would caution against statistically removing these effects, given that they might reflect the biological mechanisms behind the formation of diversity gradients, rather than being an artifact of how the distances are technically measured*”.

Referee #2

I have just a few very minor comments left:

1. The methods don't list anything about whether the climate variables were standardized prior to or during the performance of the PCA. I assume they were, as PCA on unstandardized

variables is not valid when the units of the input axes differ, but it's not listed. Please check and include.

The methods specify that “*we computed the principal components **on the correlation matrix** of these averaged variables*” (Line 476). When we perform a Principal Component Analysis (PCA) on a correlation matrix, we are effectively standardizing the variables as part of the process.

Referee #2

2. The procedure involves multiple steps of applying a grid. The first step is geographical aggregation of climate variables. It says that climate was extracted at 110 x 110 km, but when was the aggregation of the input data done (prior or after the PCA?) and how were they aggregated (centroid, mean?)?

We clarified this detail in the line 468: “*To establish each axis, we first averaged multiple current climatic variables within each spatial grid cell of approximately 110 km resolution [...]. Subsequently, we computed the principal components on the correlation matrix of these averaged variables*”. Once again, we thank the reviewer for the consistent dedication of time and expertise in the revision of our paper.

Referee #3

Thank you for thoughtfully addressing my concerns from the previous rounds. I am satisfied with the revisions made, and I am convinced that this manuscript will make a valuable contribution to the field. It advances current research on multiple fronts. One notable aspect is the introduction of a new methodological strategy, namely the examination of diversity gradients in both the geographic and climatic space. The authors have applied this strategy in a compelling manner to tackle one of the classic and long-standing questions in biology, which pertains to the causes of biodiversity gradients. The results of this study present novel and non-trivial conclusions, revealing that biodiversity gradients are influenced by both climate and climatic isolation. This finding holds meaningful implications for various research areas, including species-area relationships, island biogeography, and

conservation biology. As a result, I believe this study will be of interest to a broad readership.

After the previous rounds of detailed revisions, I do not have any substantial recommendations or suggestions. The statistical analyses are competently executed, as previously, statistical uncertainty, confidence intervals and p-values are properly reported, and results adhere to the standards in the field. In my assessment, the manuscript is well-developed and the conclusions compelling. I have been asked to specifically comment on some of the technical and statistical aspects raised by Reviewer #2. Hopefully, my feedback here will be of use to the authors and will help to address some of the concerns of the reviewer.

(1) Distances among grid cells. The authors use two approaches, where they measure distance between climate fragments and the pair-wise distances between grid cells. Both approaches seem to support the same qualitative results. Perhaps the authors could suggest that future studies use both approaches and compare the results, given that neither approach seems inherently better than the other. Reviewer #2 highlights the possibility that isolation tends to be greater in the tropics. I agree that the longitudinal distances are greater near the equator, however, the authors use the geodesic distance (shortest distance between two points on the surface of the Earth). Technically, the geodesic distances are inherently equivalent around the globe, regardless of where the two points are positioned on Earth. When comparing points between regions with tropical climates, we deliberately choose regions where the biotas have similar climatic adaptations, hence inserting biological knowledge into the analysis. Consequently, the distances will be greater in the tropics, but I would not interpret this as a methodological issue. The fact that regions with tropical climates are large and geographically spread out is arguably one of the reasons for high tropical diversity, which has biological causes (e.g. area for speciation and dispersal barriers leading to allopatry tend to be greater in the tropics). I would caution against statistically removing these effects, given that they might reflect the biological mechanisms behind the

formation of diversity gradients, rather than being an artifact of how the distances are technically measured.

We are once again thankful for the constructive comments provided by the referee, and for their talent in elegantly distilling the manuscript's main points into simple terms. We are especially appreciative of the time they dedicated to addressing the issue of isolation raised by referee #2 in the previous revision. We fully concur with the referee's viewpoint on this matter and have no further comments.

Referee #3

(2) GAM models. I also agree with Reviewer #2 that the R^2 values (% of explained variance) are hard to interpret when using GAM with the Poisson error structure. One way to circumvent the issue would be to use the fitted GAM models for predicting the observed values, then correlating the predicted values (predicted diversity, extracted from GAM) with the observed values (empirical diversity). The resultant correlation (Pearson's correlation coefficient), when squared, corresponds to R^2 . These values are easily interpretable and well-known from the linear regression, while the approach is widely used and has good precedent in biodiversity studies (e.g. Jetz and Fine 2012, Plos Biology, Figure 2). The authors could compare these R^2 values with GAM results on % of explained null deviance. I also agree that one would expect the effects of climate and climate structure to be correlated, given the wide overlap between their effects (Figure 4 in the main text). Perhaps this could be reconciled by explicitly referring to partial correlations (e.g. what climate explains) and marginal correlations (e.g. what climate explains after accounting for the effects of climate structure). This could explain why the marginal effects are uncorrelated, even though the effects of climate and climate structure largely overlap. These technical issues seem to be mostly minor, in my assessment, and easy to address, using supplementary analyses.

We appreciate the reviewer's additional suggestion regarding the calculation of the predicted R^2 . We implemented this method across all primary and supplementary analyses, and found that the results corresponded with the previously calculated adjusted R^2 . As the predicted R^2 is identical to the adjusted R^2 , we chose to maintain the presentation of the

percentage of null deviance and adjusted R^2 in the main text. However, we clearly state in the methods section that we utilized these three different R^2 computation methods and that the conclusions derived from all are consistent (*Lines 568-580*). The predicted R^2 suggested by the reviewers is now presented in all tables.

With regard to the reviewer's mention of partial correlations, we agree that this approach has value and believe we have already addressed it in our results. Figure 3 of the manuscript presents the relationship between the partial residuals (i.e., richness not explained by other predictors in the multivariate model) and each predictor. These graphs effectively reveal the influence of each variable, controlled for the effects of the others. We have intentionally not provided a measure of goodness of fit for each of these graphs to avoid potential reader confusion, as these graphs differ from a formal exploration of the importance of each predictor and their joint contribution, which is illustrated in Figure 4. Additionally, the graphs sufficiently demonstrate the form and strength of each predictor's relationship when controlling for the influence of other predictors. Consequently, one can observe that when the influence of other predictors is controlled, the target variable exhibits a unique, non-random effect that is solely dictated by that variable and not influenced by other climate or geographic components.

We again reiterated that we are grateful for the time and dedication of referee #3.